# Chitosan as an Underrated Polymer in Modern Tissue Engineering

**DOI:** 10.3390/nano11113019

**Published:** 2021-11-10

**Authors:** Marta Kołodziejska, Kamila Jankowska, Marta Klak, Michał Wszoła

**Affiliations:** 1Foundation of Research and Science Development, 01-793 Warsaw, Poland; marta.kolodziejska@fundacjabirn.pl (M.K.); kamila.jankowska@fundacjabirn.pl (K.J.); michal.wszola@fundacjabirn.pl (M.W.); 2Polbionica Ltd., 01-793 Warsaw, Poland

**Keywords:** chitosan, biopolymer, biomedicine, bioprinting, 3D, bio-ink

## Abstract

Chitosan is one of the most well-known and characterized materials applied in tissue engineering. Due to its unique chemical, biological and physical properties chitosan is frequently used as the main component in a variety of biomaterials such as membranes, scaffolds, drug carriers, hydrogels and, lastly, as a component of bio-ink dedicated to medical applications. Chitosan’s chemical structure and presence of active chemical groups allow for modification for tailoring material to meet specific requirements according to intended use such as adequate endurance, mechanical properties or biodegradability time. Chitosan can be blended with natural (gelatin, hyaluronic acid, collagen, silk, alginate, agarose, starch, cellulose, carbon nanotubes, natural rubber latex, κ-carrageenan) and synthetic (PVA, PEO, PVP, PNIPPAm PCL, PLA, PLLA, PAA) polymers as well as with other promising materials such as aloe vera, silica, MMt and many more. Chitosan has several derivates: carboxymethylated, acylated, quaternary ammonium, thiolated, and grafted chitosan. Its versatility and comprehensiveness are confirming by further chitosan utilization as a leading constituent of innovative bio-inks applied for tissue engineering. This review examines all the aspects described above, as well as is focusing on a novel application of chitosan and its modifications, including the 3D bioprinting technique which shows great potential among other techniques applied to biomaterials fabrication.

## 1. Introduction

In line with the topical coverage on the sustainable economy, the exploration of naturally-derived and renewable biopolymers, instead of fossil-fuel-based plastics, for various products’ fabrication has received tremendous attention. Biomass from marine, woody, and agricultural residuals, the most abundant renewable feedstocks on the earth, has shown an up-and-coming potential as alternative to fossil resources [1]. Liu et al. [2] listed the most important biopolymers that have so far been successfully adopted for biomaterials or bio-ink preparation. These include cellulose, sodium alginate, starch, poly(lactic acid) (PLA), agar, chitosan, and their modified derivatives and composites. As the precursor of chitosan, chitin is the most widely performing biopolymer in nature after cellulose [3].

Chitosan as a biopolymer has several useful physicochemical properties such as solubility, reactivity, adsorption, and crystallinity. In turn, its biological properties include biodegradability, antimicrobial activity, cytocompatibility, lack of toxicity, and fungicidal effects. Furthermore, the advantage of showing anti-cholestemic activity, antioxidant activity, macrophage activation, anti-inflammatory activity, angiogenesis stimulation, muco-adhesion, antitumor, granulation, scar formulation, hemostatic action and wound healing stimulation makes it a tremendously good candidate for numerous applications in biomedicine [4,5,6,7]. The role of chitosan in the pharmaceutical industry has been extensively explored. The exemplary use of chitosan in the field of medicine includes cartilage repair, bone tissue engineering, liver tissue engineering, vascular tissue engineering, drug delivery, wound healing, regenerative medicine, gene therapy, biosensing, excipient for tablets, controlled release dosage, absorption enhancer, developing micro/nanoparticles, dental implants, dental plaques, and dentifrices. Other potential possibilities of CS (chitosan) usage are in food, chemicals, cosmetics, water treatment, metal extraction and recovery, biochemical, agricultural, and environmental uses [8,9]. Consequently, chitosan has approval for application as a biomaterial by the Food and Drug Administration (FDA) in the USA and regulatory bodies of other countries and is currently used in a number of commercial products from hemostatic bandages to dietary aides [10].

A major disadvantage that limits the widespread application of chitosan in a living system is that it is not soluble in aqueous solutions. This is due to extra-molecular hydrogen bonds forming a rigid crystal structure. However, it dissolves in acid solutions at a pH not greater than 6. Acetic acid is the most commonly used acid for this purpose, though many other acids that occur naturally in the human body e.g., HCl, lactic acid, citric acid, and pyruvic acid, can also solubilize chitosan in water [2]. Chemical modifications of chitosan are alternatives to improve the properties to be applied in a broader field. The most important functional groups able to modify chitosan are the amine groups (–NH_2_). The resulting hydrogels show different strengths and weaknesses, as well as other properties [11]. Although several examples of chitosan derivatives have been used in biomedical areas, only a few, including carboxymethylated chitosan, trimethylated chitosan and polyethylene glycol (PEG), have achieved a well-established and potentially characterized application profile. These chemical modifications produce many kinds of chitosan derivatives that have sustained-release properties and are nontoxic, biocompatible, and biodegradable, and consequently widen its applications [5].

Two major CS values which determine a large part of the properties of CS are deacetylation degree (DD) and molecular weight (MW). The presence of the portable amino group along D-glucosamine residues allows elucidation of most of the chitosan properties. Indeed, the muco-adhesion of chitosan can be explained by the presence of negatively charged residues and is directly related to the DD of chitosan. If chitosan DD increases, the number of positive charges also increases, which leads to improved mucoadhesive properties. As another example, the hemostatic activity of chitosan can be related to the presence of positive charges on the chitosan backbone. Due to these positive charges, chitosan can also interact with the negative part of the cell’s membrane, which can lead to reorganization and an opening of the tight junction proteins, explaining the permeation enhancing property of this polysaccharide [5].

3D printing and bioprinting fields have experienced rapid development in recent years. The arrival of this bio-fabrication technology to tissue engineering applications has revolutionized the field, since highly complex and biomimetic constructs can be synthetized. Additionally, 3D bioprinting can generate patient-specific scaffolds with highly complex geometries while hosting cells and bioactive agents to accelerate tissue regeneration. A combination of different components such as polymers, cells, and additives are considered according to the target tissue and the 3D bioprinter used. Chitosan plays an important role in the field of bio-inks used in 3D bioprinting [12,13]. Bio-ink is the type of material used for many applications in 3D printed scaffolds. In this regard, printability, fidelity, viscoelasticity, shear-thinning, yield stress, creep, shelf life, crosslinking time, and cost are some of the essential parameters associated with the selection of bio-inks. Chitosan is one of the best 3D printing candidates due to its desirable physicochemical properties and essential features for cell adhesion, Extracellular Matrix (ECM) deposition, and finally tissue regeneration. However, the 3D printing of chitosan-based hydrogels is still under early exploration. The combination of chitosan-based hydrogels and 3D printing holds much promise in the development of next-generation biomedical implants [14].

This paper reviews the state-of-the-art in the usage of natural-derived chitosan biopolymer, its derivatives, blends with other natural polymers, synthetic polymers, and other compelling components for creating biomaterials and bio-inks for 3D bioprinting. First, we outline biological, physical and chemical chitosan properties that are key for different targeting application areas. Finally, we summarize all the latest possible uses of chitosan as a bio-ink.

## 2. Physical and Chemical Properties

Chemical, physical, mechanical, and biological properties are determined by structural parameters such as molecular mass (MW), degree of deacetylation (DD), size of polymer chains, and chitosan composition which means the distribution of two chitosan residues generating its chain. Chemical properties are determined by the linear nature of the polysaccharide, reactive amino groups, available reactive hydroxyl groups, and the ability to chelate transition metal ions [15]. The two parameters most often mentioned against a backdrop of determining chitosan properties are DD and MW. Crystallinity, viscosity in the water environment, amount of undissolved impurity particles, dry matter content and ash content are secondary analyzed parameters. Described properties depend strictly on the origin, deacetylation method, and extraction order of the chitin, depending on which specific chitosan can be obtained after the deacetylation process [16].

### 2.1. Obtaining Chitosan

Crab and shrimp shell wastes are the primary sources of biomass for the industrial production of chitin and chitosan [17]. Chitosan can also be produced from insects and the bone plates of squids and microorganisms. In fungal cell walls, chitosan exists in two forms, as free chitosan and covalently bounded to β-glucan. Chitosan is the deacetylated derivative of chitin, which is chemically defined as a copolymer of α-(1→4) glucosamine (C_6_H_11_O_4_N)_n_, having a various number of N-acetyl groups. Chitosan extraction from chitin consists of four general steps such as demineralization, deproteinization, decolorization and N-deacetylation

### 2.2. Degree of Deacetylation

The deacetylation degree is the principal structural parameter used in the context of chitosan. The difference between chitin and chitosan rests on the degree of deacetylation (DD). DD is defined as the ratio of the number of glucosamine groups to the total number of the N-acetylglucosamine (GlcNAc) and glucosamine (GtcN) groups. Deacetylated chitin can be called chitosan when the deacetylation degree is greater than 50% [18]. Other publications report values of 40% as well as 60% [19]. However, most often the DD of chitosan varies between 60–98%. A deacetylation degree of 55–70% is defined as a low deacetylated degree of chitosan, which is almost completely insoluble in water (applicable in the agro-industries textiles, fibers, and paper production technology). A deacetylation degree of 70–85% is the middle deacetylation degree of chitosan, which may be partly dissolved in water (food supplements, cosmetics, and water purification). Finally, 85–95% is a high deacetylation degree of chitosan, which has good solubility in water (biomedical and pharmaceutical industries), and 95–100% is the ultra-high deacetylation degree of chitosan and is difficult to achieve [20]. The deacetylation degree is one of the two most important descriptive parameters of chitosan and chitin. DD influences the functional property of chitosan [21]. Prior re-searches have reported that chitosan with a higher DD demonstrates stronger biological effects as well as increased water solubility. On the other hand, in most cases, the increase in the DD resulted from an increment in the mechanical properties such as tensile strength and elastic modulus [22]. Hence, it is crucial to define it. The deacetylation process is usually carried out in the solid state and results in an irregular structure due to the semicrystalline nature of the starting polymer. Along with the increasing duration of the deacetylation process, DD increases while Mw, along with the viscosity, decreases [23]. Dependences between the increase in DD and a range of physical, chemical, and biological properties are introduced in Figure 1 and Figure 2. The influence of DD on biological properties is presented by Figure 3.

### 2.3. Molecular Weight

Besides DD, molecular weight (MW) is another essential parameter that influences the bioactivity of chitosan. Like DD, lower MW chitosan (e.g., <20 kDa) usually shows more significant bioactivity than higher MW chitosan (e.g., >120 kDa). The known chitosan polymers differ in their molecular weight (from 50 to 2000 kDa). MW is strongly related to the viscosity of chitosan dissolved in dilute acids, but MW is sometimes not directly related to viscosity. It is then caused by the presence of colloidal particles in the solution [24]. MW of native chitin usually reaches a value of over 1,000,000 Da, while commercial chitosan varies between 100,000–1,200,000 Da. Lower values are caused by the possible degradation of the chitosan product under difficult conditions during the deacetylation process. The chitosan sources are also crucial for material properties; for instance, chitosan obtained from squid usually has higher molecular weight compared to chitosan produced from shellfish exoskeletons [25]. It is extremely important to determine the MW of the chitosan especially when it comes to the field of medicine [26]. In medical and agricultural domains, low-molecular-weight chitosans are preferred [27].

### 2.4. Solubility

Solid chitosan as a dehydrated material is a semi-crystalline polymer with a white or slightly yellowish color [28]. The critical DD value of chitosan for achieving insolubility in an acidic environment is over 60% [29,30]. Solubility depends on the charge density, which is determined by structural parameters such as chain length, ion concentration, degree of deacetylation, PH, the solvent nature used for protonation, the degradation of glucosamine N-acetyl units, and isolating and drying of the polysaccharide conditions. Intra-chain hydrogen bonds, which include hydroxyl groups, also influence chitosan solubility [31]. Chitosan does not dissolve in water, alkalis, or organic solvents. To omit this inconvenience, polysaccharide can be treated with an acid solution with a PH lower than its pKa, i.e., below 6.3. At such PH, chitosan is a polycation and at a PH below 4, it is fully protonated. This is due to the acid protonation of amino groups from glucosamine units. The implications of this process are electrostatic repulsions between NH+ groups such as hydrogen bonds and hydrophobic influence, which destroy interchain interactions and allow dissolving. The most common acids used to dissolve chitosan are acetic and formic acids. Other popular solvents are inorganic acids, such as nitric, hydrochloric, perchloric, and phosphoric acids [32]. In the presence of high ionic forces, the solubility of chitosan is reduced. Strong acids, such as hydrochloric acid above 0.1 M, cause chitosan precipitation due to the high proton concentration which leads to electrostatic interactions between the polymer chains. The study of chitosan protonation in the presence of 0.1 M, 1% acetic acid, and hydrochloric acid as solvents confirms that the degree of ionization depends on the PH and pKa of the acid [29].

### 2.5. Viscosity

The viscosity of chitosan in an aqueous acid solution is determined by factors such as the degree of deacetylation, ionic strength, molecular weight, concentration of chitosan, PH, and temperature of the solvent [31]. Along with increasing solvent temperature, decreasing viscosity is observed. Nevertheless, this process can be disturbed by changing the PH of the solution, depending on the acid used as a solvent. For example, the viscosity of chitosan dissolved in aqueous acetic acid increases with decreasing PH. Chitosan dissolved in dilute acids behaves like a non-draining worm-like molecule, the molecular configuration of which depends on ionic interactions between poly-ion-counterions [32].

### 2.6. Coagulation

Chitosan shows excellent coagulation and flocculation properties. Chitosan is capable of interacting with negatively molecules of charged proteins, polymers, solids, and dyes due to the high density of amino groups. In the presence of metal ions such as nitrogen, the amino groups of chitosan act as an electron donor responsible for the selective chelation of metal ions [21].

## 3. Biological Properties

Chitosan as a natural biopolymer has become an attractive biomaterial due to the extraordinary biological properties that allow it to be used in a number of biomedical and pharmaceutical fields. The crucial chitosan properties are biocompatibility, non-toxicity, and biodegradability. Clinical tests have shown that chitosan-based biomaterials do not cause any allergic reactions or inflammations [33]. In addition, chitosan shows antimicrobial [34,35], antioxidant [36], antiallergic [37], anti-inflammatory [38], anticoagulant [39,40] antioxidant [41], hemostatic, antitumor due to macrophage activation [42], analgesic, and hypo-cholesterolemic properties [43]. Chitosan also has a positive effect on the human immunodeficiency virus (HIV) [44], inhibition of Alzheimer’s disease solution [45], and adipogenesis [46]. Chitosan according to Ramana et al. [47] can work as a chitosan-based nano-delivery strategy for saquinavir. The polycationic nature of chitosan is expected to favor the deposition of the complement proteins on the nanoparticles, resulting in their uptake by the macrophages through complement receptors. The macrophages serve as HIV-1 reservoirs, thus efficient drug delivery to the cells via chitosan could be an added advantage. The drug release from the chitosan carriers could be accentuated in the acidic endosome due to extensive protonation and destabilization of the polymer matrix. In the end, the loaded chitosan nanoparticles were found to exhibit superior potency than the free drug even in nanogram levels. Both strains of HIV-NL4-3, and Indie-C1-were found to respond to chitosan containing saquinavir, indicating the potency of this system as an effective anti-HIV system. Chitosan enables three-dimensional tissue growth thanks to its non-protein matrix [48]. Chitosan is also widely used as a composition material in drug carrier systems. Chitosan popularity is due to its immunoenhancing effect which is beneficial for patients [42]. It has the ability to stimulate cell proliferation and can act as a biological primer for the proliferation and reconstruction of cellular tissue [49]. According to Malette et al. [48] chitosan is defined as a polysaccharide that is structurally similar to extracellular glycoprotein carbohydrate’s ability to perform similar morphogenetic functions.

### 3.1. Biocompatibility

Biocompatibility is a crucial chitosan property that is required in biomedical fields [50]. Tomihata et al. [51] proved the correlation between DD and biocompatibility. The research was performed on a rat implant model using chitosan films. Materials containing chitosan with a DD of 31% induced stronger inflammatory reactions compared to materials containing chitosan with a lower DD and exhibited a slower rate of degradation. The impact of DD on the degradation rate was confirmed by Barbosa et al. [52]. In their work two different DD chitosans, 4% and 15%, were contained in porous scaffolds using a subcutaneous air-pouch inflammation model. Chitosan 15% DD caused a significantly higher level of neutrophilia and increased adhesion of inflammatory cells in the early phase of implantation. Chitosan 4% DD caused a significantly lower reaction present in the inflammatory exudate leukocytes. Moreover, chitosan 15% DD caused a rise in the thickness of the collagen capsule as well as a strong infiltration of inflammatory cells within the implantation area. Inflammation and healing are closely related processes, hence the degree of DD of chitosan should be considered when creating materials in tissue engineering for tissue repair and regeneration.

### 3.2. Biodegradation

Chitosan seems to be degraded in vivo by a series of unspecific enzymes, but mainly lysozyme, present in all mammalian tissues, was reported to have such a property. In vitro degradations of chitosan occur via oxidation, chemical, or enzymatic hydrolysis. Biodegradation of chitosan leads to the release of amino-saccharides, which are incorporated into glycosaminoglycan and glycoprotein metabolic pathways or excreted [53]. It was observed by Zhang et al. that a link occurs between a degradation rate and molecular mass, the distribution of N-acetyl D-glucosamine residues, deacetylation degree, and consequently on crystallinity [54]. The biodegradation rate increases when crystallinity decreases. The smaller chitosan chains are more efficiently biodegraded than higher molecular mass chitosans. Su et al. carried out a study of in vivo degradation of carboxymethyl chitosan (CMCS, 20% wt.%). The hydrogel was injected into the back of the rats at room temperature. The majority of the gels disappeared within 10 days after the injection. CMCS hydrogels were completely degraded and resorbed at 19 days [55].

### 3.3. Cytocompability

A great number of publications report chitosan cytocompatibility. Keratocytes [56], chondrocytes [57], osteoblasts [58], hepatocytes [59], and Schwann cells [60] are examples of cell types that have been successfully grown on 2D and 3D chitosan materials. Amaral et al. [57] highlight an association between DD and the degree of cytocompatibility that increases with decreasing DD. Cytocompatibility is coupled to cell adhesion, hence the conclusion that lower DDs promote cell adhesion. The same authors hypothesized that DDs influence cell adhesion and osteoblast differentiation by affecting the adsorbed layer of adhesion proteins. On completion of the experiments with the 125I-fibronectin protein, it was hypothesized that protonated amino groups derived from glucosamine units are able to modulate cell adhesion to chitosan by promoting the adsorption of adhesion proteins to the cell’s membrane, such as fibronectin.

### 3.4. Antimicrobial and Antifungal Properties

Antimicrobial properties are strictly correlated with chemical, physical and biological factors. Molecular weight and degree of deacetylation are two decisive parameters against the backdrop of antimicrobial properties. Chitosan with a higher degree of deacetylation possess better antimicrobial properties. This effect consists of an increase in electrostatic bonding to the cell membrane which yields to a permeabilization effect. Chitosan with a lower molecular mass ratio is characterized by higher antibacterial activity which is directly correlated with the ability of chitosan molecules to penetrate the cell membrane and accumulate in the bacterial cell [60]. To determine antimicrobial properties, other factors such as PH, the salinity of the medium, temperature, content of divalent cations, applied solvents, type of suspension medium, and the growth phase of microorganisms need to be taken into account. According to Regiel et al. [61], chitosan exhibits antibacterial activity against both Gram-positive and Gram-negative bacteria. Chitosan efficiency on a variety of microorganisms has been proven in research conducted on a large number of bacterial strains, fungi, and yeasts. In turn, Hong Kyoon No et al. [62] describe significant growth inhibition of most of the tested bacteria, such as *Escherichia coli*, *Pseudomonas fluorescens*, *Salmonella typhimurium*, *Vibrio parahaemolyticus*, *Listeria monocytogenes*, *Bacillus megaterium*, *Bacillus cereus*, *Staphylococcus aureus*, *Lactobacillus plantarum*, *Lactobacillus brevis*, *Lactobacillus bulgaricus*. Seo et al. [63] tested the effect of chitosan on the growth of 11 different bacteria and found that the chitosan MIC90 (Minimum Inhibit Concentration; minimum concentration of the sample that is needed to inhibit 90% of the fungus colonies) ranged from 10 to 1000 ppm. Of the organisms tested, growths of *E. coli*, *P. fluorescens*, *B. cereus*, and *S. aureus* were inhibited by chitosan concentrations of 20, 500, 1000 ppm (that is, 0.002%, 0.05%, 0.1%), respectively. Uchida et al. [64] reported the chitosan MIC for *E. coli* and *S. aureus* to be 0.025% and 0.05%, respectively. According to Hong Kyoon et al. The MIC values of chitosan ranged from 0.006% to 0.03% except for *P. aeruginosa* (0.05%) and the food-borne pathogen *S. typhimurium* (more than 0.1%). The latter was the most resistant bacteria strain studied [62]. Moreover, it was shown that the growth inhibitory effect is specific for the type of bacteria strain. Chitosan showed a stronger antibacterial activity against Gram-positive bacteria (e.g., *S. aureus*) compared to Gram-negative bacteria (e.g., *E. coli*). Trong-Ming Don et al. [65] reported that the antimicrobial activity of pure chitosan was effective against *S. aureus* by inhibiting bacterial growth by more than 90%. This would indicate that, by using only a small amount of chitosan (chitosan concentration 0.1%, a satisfactory antimicrobial effect could be achieved.

The five main mechanisms of the antimicrobial activity of chitosan have been distinguished. These include: (i) the formation of an impermeable layer that prevents the transport of essential solutes, (ii) inhibition of RNA and protein synthesis by permeation into the cell nucleus (iii) chelation of nutrients (iv) interaction with flocculated proteins and (v) disturbance of filamentous fungi membrane [29].

A significant amount of research has been completed on fungi as organisms susceptible to chitosan. Chitosan in its free polymer form has been proved to have antifungal activity against *Aspergillus niger*, *Alternaria alternata*, *Rhizopus oryzae*, *Phomopsis asparagi*, and *Rhizopus stolonifera*. However, the results obtained by Palemira-de-Oliveira et al. [66] and Garcia-Rincon et al. [67] report unambiguous results showing the antifungal activity of chitosan. The studies were carried out on various *Candida species* and *E. stolonifer*, on low, medium, and high molecular weight chitosan. The results showed that the fungi are dependent on the concentration of chitosan in the solution. The results confirm that antifungal properties rely on the destabilization of the physiological function of the cell membrane. However, the chitosan molecular weight has no significant impact on the properties previously described. Whereas, LMW (Low Molecular Weight) chitosan is more effective in inhibiting mycelial growth, HMW (High Molecular Weight) chitosan has a greater impact on the inhibition of spore development [68].

### 3.5. Antioxidant Activity

The antioxidant activity of chitosan consists of eliminating reactive oxygen species (ROS) and repairing damages caused by them. This activity has been tested in vitro and in vivo. Liu [69] observed that the addition of 0.02% chitosan had a positive antioxidant effect, but to a lesser degree than ascorbic acid. A slight increase in the concentration of chitosan increased the antioxidant properties of the level of vitamin C. Then chitosan was able to reduce serum free fatty acid and malondialdehyde (MDA) concentrations. It also could increase antioxidant enzymes activities such as superoxide dismutase (SOD), catalase (CAT), and glutathione peroxidase (GSH-PX). In conclusion, chitosan could regulate the activity of antioxidant enzymes and reduce lipid peroxidation [70].

Three mechanisms of the antioxidant activity of chitosan have been distinguished: (i) proton donation or electron transfer from NH_2_ groups, (ii) proton donation from OH groups at C2, C3, and (iii) C6 positions, and chelation of ferrous ions [16]. A significant number of studies have been completed on the relationship between the molecular weight of chitosan and its antioxidant properties [71]. Based on seven different chitosan samples with different molecular weights (2.8–931 kDa) Tomida et al. [72] proved that chitosan with the lowest molecular weight was the most effective in preventing the formation of carbonyl groups in plasma proteins, which may be exposed to proxy radicals which have an impact reduction of oxidative stress. Moreover, Feng et al. [73] reported that the chitosan with the lowest molecular weight (2.1 kDa) shows the highest antioxidant activity.

The case of degree of deacetylation is slightly different. There is no direct correlation between DD and antioxidant activity. However, according to Yen et al. [74], the longer the deacetylation process time, the higher the deacetylation degree and thus the greater the number of amino groups on the C2 carbon, which increased their antioxidant properties. All crab chitosans exhibited showed moderate to high antioxidant activities of 58.3–70.2% 1 mg/mL and high antioxidant activities of 79.9–85.2% at 10 mg/mL.

### 3.6. Enzymatic Degradation

Chitosan can be degraded by enzymes that have the prospect to hydrolyze glucosamine-glucosamine, glucosamine-N-acetyl-glucosamine, and N-acetyl-glucosamine-N-acetyl-glucosamine units [75,76]. Enzymes involved in the hydrolysis of chitosan are partially unidentified. Deacylated chitosan is partially depolymerized by various types of lysozymes present in serum, saliva, other fluids [77], and bacterial enzymes present in the colon [78]. The research conducted by Loončarević et al. [79] obtained data proving a strong relationship between the activity of lysozymes and the PH value of the used medium (activity under pH 6.0–9.0), which ultimately determined the weight loss of chitosan.

As DD increases, the initial rate of degradation increases. The lowest degradation rates are attributed to chitosan with low DD and a high level of crystallinity and intermolecular bonds [80].

### 3.7. Immunoadjuvancy

Chitosan acquires chemotactic properties that determine the ability to respond to chemical stimuli such as neutrophils. Moreover, it exhibits the biological ability to activate macrophages to tumor activity in order to produce interleukin-1 and nitric oxide [44].

### 3.8. Absorption-Enhancing Effect

The effect described in the subtitle is based on the ability of chitosan to open tight epithelial junctions. Chitosan owes the effect of absorbing interactions to both its positively charged residues and the cell membrane, which in turn results in the reorganization of the tight junction-associated proteins [81,82]. According to Aranaz et al. [16], it is possible to open the epithelial junctions only when the chitosan DD reaches 80%. The investigation conducted by Mei et al. [83] confirmed the effect of molecular weight by comparing the absorption of 2,3,5,6-Tetramethylpyrazine by different Mw chitosan (400, 200, 100, and 50 kDa). Their results showed that with an increase Mw the level of absorption also increases, but only to the value of 100 kDa. For higher molecular weight ratios, no significant difference was observed between the absorption levels.

### 3.9. Immunomodulatory (Anti-Inflammatory Action, Analgesic) Properties

A very interesting property of chitosan is its immuno-stimulating effect. This relies on stimulating innate immune cells to release a wide spectrum of pro-inflammatory and anti-inflammatory cytokines, growth factors, bioactive lipids, and chemokines [84]. Despite many years of research on its immunostimulatory properties, the exact pathways of the action are yet to be understood. Described properties depend on the DD, MW poly-dispersion, chitosan dose, and the presence of pro-inflammatory co-stimulation agents [58,85]. Bradykinin is a polypeptide that is one of the major mediators of inflammation and pain [86]. Kim et al. [87] in their research proved that the level of bradykinin is reduced as a result of chitosan action. Furthermore, Huang et al. [88] conducted research on scalded rats. He proved that, at day 28 of wound healing, re-epithelization was completed in 3% chitosan and 3% carboxymethyl chitosan groups, while only partial re-epithelization was found in control (sterile saline).

### 3.10. Hemostatic and Blood Clotting Properties

Hemostasis is the process that occurs in the first stage of a wound injury. As a result, coagulation occurs, which is responsible for the bleeding control process and wound closure [89]. Chitosan shows very strong hemostatic properties even when the patient is administered anticoagulants [90]. Thanks to research by Fong et al. [85] carried out on chitosans with 80 to 98% DD and 3 to 400 kDa MW, it can be concluded that chitosan, and in particular dressings made with its participation, have fantastic hemostatic properties that shorten the hemostatic time and effectively inhibit blood loss. The procoagulant activity of chitosan depends on its effectiveness in reducing antithrombin production.

### 3.11. Antitumor Activity

The anti-cancer properties of chitosan are related to DD, MW, the source of chitosan, and the PH of the solution in which it is located [91]. So far, there is no clear answer as to what type of chitosan has the strongest anti-cancer properties. In vivo studies by Fong et al. [85] suggest that the antitumor effect increases with increasing MW. This may be due to diffusion constraints caused by the high viscosity of HMW chitosan samples. However, viewed from another angle, LMW chitosan samples can penetrate easily through the cell membrane by dint of its low viscosity, which in turn promotes inhibition of cell lines [92].

### 3.12. Macrophage Activation

According to Peluso et al. [93], chitosan has a stimulatory effect on macrophages and the macrophage activation is mainly attributable to the NAGA (N-Acetylglucosamine) unit of the chitosan molecule rather than to the glucosamine residue. Research conducted by Mori et al. [94] confirms that the death of the peritoneal macrophages can be inducted by chitosan. Resident peritoneal macrophages were prepared from BALB/c mice and treated with chitosan (50 mg/mL). After 24-h the cells had aggregated around the polysaccharide and died whereas the cells from the control group were unaffected. After 12-h of exposure, chitosan induced extensive death of the cells (58.3% compared to 19.4% in the control).

### 3.13. Pro-Health Properties (Antihyperlipidemic, Mineral Absorption, Bodyweight Reduction)

Chitosan as a dietary substance has several valuable properties. Some of the crucial abilities are antihyperlipidemic, mineral absorption, and bodyweight reduction. Studies conducted on Sprague-Dawley rats demonstrated the potential for lowering cholesterol levels in both animals and humans. More specifically, chitosan not only enables lowering the level of total concentration (TC) cholesterol but also reduces the level of low-density lipoprotein LDL (Low Density Lipoprotein)cholesterol while increasing HDL cholesterol. Chitosan was first shown to reduce serum cholesterol in humans in 1993 [95] when adult males fed chitosan-containing biscuits for two weeks (3 g/d for the first week 1.6 g/d for the second week) experienced a significant decrease of 6% in total cholesterol. Additionally, two studies have reported serum cholesterol reductions with chitosan treatment. Obese women consuming 1.2 g of microcrystalline chitosan for 8 weeks demonstrated significant reductions in LDL, although not total serum cholesterol [96]. Most recently, female subjects with mild to moderate hypercholesterolemia receiving 1.2 g of chitosan per day experienced a significant decrease in total serum cholesterol [97]. Panith et al. [98] presented the cholesterol mechanism of action, which is based on cholesterol, and bile salts binding, as well as the relationship with MW and the origin of chitosan. For the chitosan to have fat binding capacity, its DD must be at least 80%. In addition, he proved that increasing MW and tap density of chitosan could significantly increase fat-binding ability. In turn, Gallaher [99] described the increasing fecal excretion of bile acids as a mechanism of cholesterol reduction. He has also shown greatly increased (approximately three-fold) fecal bile acid excretion in rats fed diets containing 7.5% chitosan. Higher MW chitosans (2100 and 890 kDa) show significantly higher fat binding capacity than 30 kDa. When tested at chitosan to fat 1:40, the ratio often used as a dietary supplement, higher tap density chitosan showed an improved fat-binding capacity at around a two-fold increase. Interestingly, 2100 kDa high density (HD1P) could maintain the highest oil entrapment, ranging between 0.77 and 27.50 g oil/g chitosan depending on the interaction ratios [100].

In animals, 20 female SHRSP (Stroke prone Spontaneously Hypertensive rats) studies showed a regulatory effect of chitosan on the Ca mineral. Improper Ca balance in the body is directly related to bone deterioration, Fe absorption, and imbalance in the bone mineral content [101]. Chitosan is responsible for increasing the level of Ca excretion in urine, which reduces the concentration of this mineral in the serum without inhibiting its absorption [102].

## 4. Chitosan Blends with Other Materials

The field of biomaterials must meet requirements such as appropriate mechanical parameters, biocompatibility, and stability in the aquatic environment [103]. To improve the properties or create new ones, chitosan can be combined with other natural or synthetic polymers. Obtaining blends by mixing polymers allows the production of components with specific, desired properties and applications. For some time now, it has been very popular to combine several natural polymers and replace them with synthetic ones [104]. There are two main methods used when combining chitosan with other polymers: mixing in solution, i.e., dissolving chitosan in a solvent, most often dilute acetic acid, then adding other polymers under continuous stirring conditions, and evaporating the solvents. To improve the mechanical properties a cross-linking agent is very often added. The resulting features of the blends strictly depend on the defined interactions between the polymer component chains such as hydrogen, ionic, dipole, π-electron bonds, and charge-transfer complexes. The second widely applied approach is melt blending, i.e., blending under melt conditions [105]. The most frequently improved properties in biomaterials by the addition of chitosan are increasing hydrophilicity, mechanical properties, antibacterial properties, and improving blood compatibility [106]. Another method of improving the characteristics of chitosan is to subject it to a wide range of modifications.

### 4.1. Natural Polymers

#### 4.1.1. Gelatin

Gelatin is a product of collagen hydrolysis and can be used as a cheaper replacement while maintaining its biocompatibility. One method of creating homogeneous mixtures is the formation of a polyelectrolyte complex between a positively charged chitosan and a negatively charged polymer such as gelatin, which is negatively charged when in a medium with a PH below its isoelectric point (4.7). The positively charged chitosan ammonium ions interact with carboxylate groups to form ampholytic gelatin [107]. Ryoung et al. [108] proved with the example of tooth cells that gelatin enables adhesion and rapid proliferation on chitosan/gelatin scaffolds. Other studies reported that gelatin activates macrophages, shows a high hemostatic effect, and does not cause antigenicity. Gelatin, due to the hydrogen and van der Waals interactions, mixes very well with chitosan in all proportions, as can be seen as no phase segregation or secondary peaks on scanning electron micrographs [109]. Both polymers have strong hydrogen bonds inside and between them, which contribute to the formation of a blend with enhanced mechanical properties compared to individual ones [110]. Kumar et al. [111] prepared chitosan-gelatin scaffolds with 4% (*w*/*v*) gelatin. Due to the addition of gelatin, the pore sizes become distinctly smaller. Chitosan-gelatin scaffolds could be more suitable for tissue engineering than chitosan scaffolds, and even more so for synthetic metallic scaffolds. The average pore size below 300 µm helps in osteoblast and cell proliferation in scaffolds. The pores in chitosan--gelatin blends were heterogeneous and perpendicular, while they were loosely packed in the chitosan sample scaffold. The pore sizes in the chitosan-gelation samples were influenced by gelation concentration. The combination of gelatin with chitosan improved the mechanical properties and biodegradation kinetics of the scaffold and formed a prominent amorphous composite. Higher tensile strength was observed in scaffolds prepared by chitosan-gelatin due to the possible formation of durable hydrogen bonds during the preparation of samples. According to Pulieri et al. the preferred blends are those with a predominant amount of gelatin, as it has a strong influence on the physicochemical properties due to increasing hydrophilicity with increasing content. The formation of amide and ester bonds was observed between the macromolecules, which influenced the thermal resistance, degree of swelling and dissolution, measurement of the contact angle, and mechanical properties. Blends containing 80 wt.% gelatins support the adhesion and proliferation of neuroblastoma cells [112]. Chitosan blends with the addition of gelatin cause much greater cell adhesion than each of the materials separately. After 24 h culture time, gelatin-rich samples (chitosan/gelatin 20/80) and (chitosan/gelatin 40/60) in the form of cast films displayed the highest cell density compared to chitosan by itself (*p* < 0.05). In addition, Pulieri et al. [112] observed the formation of blood vessels around the chitosan/gelatin samples (angiogenesis).

#### 4.1.2. Collagen

Collagen is a protein widely used in tissue engineering. This is due to its excellent biocompatibility, biodegradability and weak antigenicity [113]. However, its poor mechanical properties and stability force it to be combined with other polymers [114]. To this day, 29 distinct collagen types have been characterized (collagen types I, II, III, V, and XI). Type I collagen is currently the gold standard in the field of biomaterials tissue-engineering [115]. Interactions between chitosan and collagen promote the formation of complexes through electrostatic interactions or hydrogen bonding. Due to the cationic nature of chitosan, it can interact with the anionic amphiphilic parts of collagen molecules [116]. Chitosan can be combined in any proportion with collagen. The addition of collagen to chitosan enables the creation of biomimetic structures similar to natural bone [117]. It creates the right environment for regeneration. The osteogenic and chondrogenic potential scaffolds obtained as a result of their combination increase the amount of calcium and sulfated glycosaminoglycans [118]. The obtained blends can be used as bioactive substances that influence the behavior of cells and the ultrastructure of collagen-chitosan composites greatly influence the microenvironment for cell growth. Moreover, they are flexible. Cell proliferation (K562 cells) on collagen-chitosan matrices was higher for proportion chitosan:collagen 1:1 than 3:1 or 4:1. When using a 50/50 weight ratio, they improve mechanical strength, with higher resistance to enzymatic degradation, and their structure shows the ability to bio-resorb after the development of new tissue. The 1:1 chitosan-collagen (12.75 ± 1.97 KPa) had significantly higher compressive strength than collagen (10.88 ± 1.59 KPa) and 1:2 chitosan-collagen (10.54 ± 1.23 KPa). In vivo studies demonstrated the chances of the obtained materials promoting nerve regeneration, osteoblast growth and differentiation, as well as muscle cell proliferation and improvement of angiogenesis. Additionally, they can be used to deliver drugs to the body. The addition of collagen to chitosan increases the porosity and the pore diameter [119].

#### 4.1.3. Hyaluronic Acid

Hyaluronic acid is a glycosaminoglycan present in the body as the main component of the cell-matrix. Due to its high degree of biocompatibility and non-cytotoxicity, it is widely used in the field of biomaterials. It also has anti-inflammatory and anti-swelling properties. Its disadvantage is a very fast rate of degradation. The miscibility of chitosan and hyaluronic acid depends mainly on the ionic strength of the solution [120]. The physical properties of the chitosan blends with hyaluronic acid show clear intermolecular interactions between the two polymers. These interactions are related to the presence of amine, hydroxyl, and sometimes carboxyl groups. Additionally, hydrogen bonding appears, which improves mechanical and thermal properties [121]. Hyaluronic acid in conjunction with chitosan and collagen is mainly used as an osteogenic material due to the increased process of mineralization, tissue regeneration, cell proliferation, and wound healing. Additionally, the obtained structures resemble the biomimetic structure of bone. The pore size, thermal and mechanical properties are appropriate for tissue engineering [122]. Deng et al. [123] investigated the formation of chitosan/hyaluronic acid hydrogel in situ. In a rat model, they showed that the attained material promotes the formation of abdominal tissue through cell infiltration and the deposition of ECM. They also observed increased angiogenesis due to the enhanced angiogenesis-related factor VEGF (Vascular Endothelial growth Factor). VEGF factor increased to above three-fold change in mRNA expression compares to the control (one-fold change).

#### 4.1.4. Alginate

Alginate is, next to chitosan and hyaluronic acid, another commonly used polysaccharide for the production of biomaterials. The materials obtained on its basis are bioactive, biocompatible, hemostatic. Thanks to electrostatic interactions and hydrogen bonds, it interacts with chitosan. The addition of sodium alginate greatly improves the mechanical properties of the obtained composites. Dialdehyde alginate can in turn be used as a cross-linking agent [124]. Iwasaki et al. also investigated the combination of chitosan and alginate. Their data identifies chitosan as supporting scaffold structure while alginate supports cell proliferation [125]. Wang et al. confirmed the above properties characterized by other authors and also reported that the obtained materials have a very good degree of nerve cell proliferation (for OECs a 149% increase and for NSCs a 230% increase). The porosities of the chitosan-alginate complex in the mentioned study were determined to be 85.6% ± 4.7%, and the scaffold had a pore size of approximately 70–150 µm, which was favorable for cell attachment and migration [126].

#### 4.1.5. Agarose

Agarose is a biocompatible polysaccharide obtained from seaweed. The main distinguishing feature of agarose from other polymers is its ability to form thermally reversible gels even at low concentrations [127]. The primary disadvantage of this polymer is the lack of cell adhesion [128]. The significant similarity of the chemical structures of chitosan and agarose make them possible to combine and investigate their properties as candidates for biomedical applications [129]. The addition of agarose to chitosan significantly slows down the decomposition process compared to the use of chitosan separately. Slight improvement in mechanical and swelling properties obtained in a study by Fan et al. suggests possible interactions between the two polymers through hydrogen bonding. The dry tensile strengths of the alginate:chitosan 9:1 and alginate:chitosan 3:7 were higher than that of pure alginate, and the maximum value was observed at 30% Carboxymethyl chitosan content which achieved 13.80 cN/tex in the dry state [130]. The FTIR ( Fourier Transformed Infrared Spectroscopy) and FESEM (Field Emission Scanning Electron Microscopy) results confirm the high miscibility of both polymers. Moreover, the obtained blend showed smooth, homogeneous surface properties compared to pure components [131]. According to Hu et al., the preferred blend of chitosan and agarose is 60 wt.%. agarose. Such composite showed much better tensile strength, elongation at break, water vapor transmission rate, water absorption capacity, and antimicrobial activity compared to pure chitosan. The tensile strength value of the films increased from 2.72 to 5.31 MPa as the agarose concentration rose from 0% to 40%. The tensile strength of the composite film with an agarose mass concentration of 40% was approximately double that of the chitosan film without any agarose. The results showed that the water vapor transmission rate values of the composite films increased as agarose was added in greater concentrations (from 800 g/m^2^d without agarose to 1200 g/m^2^d for 80% agarose) [132]. Moreover, according to Zamora-Mora, chitosan-agarose gels show a higher elastic modulus than agarose gels. The elastic modulus of the composite gels increased with agarose concentration, reaching the value of 1 kPa at 20 °C for chitosan/agarose-2 (2% *w/v* agarose). In addition, composite gels had a higher elastic modulus than agarose gels (G′ = 824 ± 10 Pa for the sample chitosan/agarose-1.5 (1.5% *w/v* agarose) and G′ = 540 ± 44 Pa for agarose-1.5) [133].

#### 4.1.6. κ-Carrageenan

Carrageenan is a natural high molecular weight carbohydrate polymer present interstitially in the plant structure of specific species of these red seaweeds. The combination of κ-carrageenan with chitosan dissolved in various types of dilute acids (acetic, lactic, citric, malt, ascorbic) was investigated as an example of a membrane formation [134]. An increase in mechanical properties was assessed when ascorbic acid was used, which may indicate an interaction between organic acids, κ-carrageenan, and chitosan. The tensile strength and the elongation of the films were found to increase. Gel films showed excellent mechanical performances, with breaking stress, breaking strain, and Young’s modulus being 2–6.7 MPa, 80–120%, and 1.2–25 MPa, respectively, which were superior to the values of most existing biopolymer-based hydrogels. The good miscibility of κ-carrageenan with chitosan is due to the presence of oppositely charged groups in both polymers. The obtained hydrogels had very good mechanical strength, better than most of the already existing polymer-based hydrogels. These features can be related to the synergistic effect of ionic and hydrogen bonds between κ-carrageenan and chitosan molecules. Additionally, the obtained blends had very good self-healing, biocompatible and anti-adhesive properties [135]. κ-carrageenan containing side sulphates can be used as an additional ionic cross-linking agent which gives a stable effect to chitosan hydrogels under acidic conditions [136].

#### 4.1.7. Silk

Silk, popularly known in the textile industry for its luster and mechanical properties, is produced by cultured silkworms. Silks are fibrous proteins synthesized in specialized epithelial cells that line glands in these organisms. Silks are generally composed of β-sheet structures due to the dominance of hydrophobic domains consisting of short side-chain amino acids in the primary sequence [137]. Kweon et al. investigated the possibility of combining an aqueous solution of chitosan in acetic acid with silk fibroins (SF) [138]. The conformation of the obtained mixture showed the β-sheet structure which significantly improved the thermal decomposition. In turn, Samal et al. [139] developed chitosan blends with silk to provide new possibilities for modulating mechanical and strength properties. The combination of chitosan with silk significantly improved the physicochemical properties of the obtained blends and additionally allowed imitation of the natural environment of some tissues. Obtained chitosan:silk (1:2) blend had twice the strength factor (120 MPa) of chitosan alone (60 MPa) [140]. Chitosan and silk scaffolds produced by the ultrasound technique showed porous interconnected structures which support cells adhesion and proliferation [139]. Eivazzadeh-Keihan et al. [141] in their research proved that the additive of silk fibroin can significantly increase the compressive strength from 65.42 kPa for chitosan alone to 649.56 kPa for the described composite additionally combined with Mg(OH)_2_.

#### 4.1.8. Starch/Pullulan

Because of its inherent biodegradability, overwhelming abundance, and annual renewability, starch is one of the most promising natural polymers. A review of the chemical shows a carbohydrate polymer consisting of anhydro-glucose units linked together primarily through α-d-(1→4) glucosidic bonds [142]. In turn, pullulan is a biopolymer produced by strains of the polymorphic fungus *Aureobasidium* pullulans as an extracellular, water-soluble polysaccharide. It is produced by using various substrates such as starch, distilled by-products, bakery waste, or agro-industrial residues [143]. According to Lazaridou et al., use of starch and pullulan can have a significant impact on the thermomechanical properties of the produced mixtures. The film sensitivity to plasticization was as follows: starch/chitosan > pullulan/chitosan > chitosan [144]. The mixture of starch and chitosan was also investigated in the context of antimicrobial properties. Zhai et al. proved that introduction of 20% chitosan to the starch blend can significantly improve not only the antibacterial properties but also the tensile strength and flexibility. The tensile strength increased by about 1. times (from 31 MPa for 100% starch to 43 MPa for 80% starch/20% chitosan). Elongation at break rose due to the increasing amount of the chitosan content (from 20% for 100% starch to 30% for 80% starch/20% chitosan). Because of the difficulties of forming homogeneous starch/chitosan mixture with higher than 20% content of chitosan, this was the maximum value used in the described experiment [145]. As also suggested by Xu et al. [146], the percentage elongation at break with the increasing proportion of starch in the blend could be boosted. Wittaya-Areekul et al. investigated the effects of corn starch, dextran and polypropylene glycol as a plasticizer. It turned out that the use of the above compounds improves physical properties such as vapor and oxygen penetration and water absorption. Moreover, propylene glycol improves flexibility [147].

#### 4.1.9. Natural Rubber Latex (NR)

Natural cis-1,4-polyisoprene rubber occurs mainly in the *Hevea brasiliensis* tree. Due to the highly stereoregular microstructure of the rotational freedom of α-methylene bonds and entanglement, it is characterized as a very good mechanical properties compound. These include elastic and damping properties as well as good elasticity. Its disadvantages are poor chemical resistance and processability. Chitosan was easily mixed with natural rubber latex by the solution casting method. The prepared samples have significantly improved mechanical properties compared to the separate components. With increasing chitosan content by 300%, modulus of the NR/chitosan bio-composite film was significantly increased from 2 MPa up to a value of 8 MPa. In addition, there is good adhesion between the formed phases [148]. Paroibut et al. obtained the best NR/chitosan blend consisting of 10% chitosan, 90% NR. The uncured films had an 11.2 MPa tensile strength and 1259% elongation at break compared to 50/50 ratio chitosan/NR 2.6 MPa and 2%, respectively [149].

#### 4.1.10. Cellulose

Cellulose can be classified according to its origin as vegetable, microbial or animal. Bacterial cellulose (BC) is most commonly used as an additive to biomaterials. It is produced by the Gram-negative bacteria *Acetobacter xylinum* and has remarkable biocompatible properties which favor its use in the field of biomedicine. Since the chemical structures of chitosan and cellulose are similar, they can be easily mixed. Cellulose in combination with chitosan is a more and more widely used blend with antibacterial, anti-odor, adsorption of metal ions, increasing water absorption, appropriate porosity structure, and self-healing properties. According to Duan et al. [150], cellulose combined with chitosan can mix due to intramolecular inclusion interactions. Such a blend gives better stability compared to polymers used separately. The self-healing properties of the blends were studied and found that the healed gel can recover 72.77% of its original compressive strength due to the host-guest interactions [151]. In another work, Wu et al. [152] found that 32% loading of cellulose in chitosan films produced by solution casting method caused 12 and 30 times improvement in the tensile strength and Young’s modulus, respectively.

#### 4.1.11. Carbon Nanotubes (CNT)

Immediately after the discovery of carbon nanotubes in 1991, they became a sought-after material in the fields of biomedicine and tissue engineering. By using it, the mechanical properties of many materials can be significantly improved. CNTs are cylindrical carbon tubes possessing nanometer diameters with significant length (4100 nm) resulting in very large aspect ratios and can adsorb a large number of proteins, which can benefit cell attachment and proliferation [153]. Cytotoxicity of nanotubes is a controversial topic that has not been clearly defined as yet. There are many types of carbon nanotube, but the most commonly used in the context of chitosan are Multi-Walled Carbon Nanotubes (MWCNT). Aryaei et al. [154] showed that even the addition of small amounts of MWCNT has a significant impact on the optical and mechanical properties and the binding energy of particles. Adding 1% by weight of MWCNT to the chitosan hydrogel can increase the tensile modulus and the tensile strength by up to 47%. Moreover, no cytotoxic effect of the added nanotubes was demonstrated. Moreover, an increase in the viscosity of the obtained materials was demonstrated. According to Arora S et al. the lowest cytotoxicity occurred in systems with a concentration of 1 μg/mL [155].

### 4.2. Synthetic Polymers

Compared to natural polymers the unquestionable advantage of synthetic polymers are better mechanical properties, thermal stability, and the ability to form a wide range of shapes and geometries. Their synthesis can be tailored to produce a specific molecular weight, chemical structure, end group chemistry, and exact composition. Another advantage of synthetic polymers is much longer biodegradation time. To increase the possibility of their wider use, they can be modified [156].

#### 4.2.1. PVA Poly(Vinyl Alcohol)

PVA is a biocompatible, non-toxic, hygroscopic, and semicrystalline polymer [157] with high chemical stability and filmmaking capability. It shows minimal cell adhesion and the ability to absorb proteins [158]. Chitosan linked with PVA gives a non-miscible system. It is caused by a stronger interaction between PVA macromolecules and interactions between PVA and chitosan. However, Khoo CGL et al. [159] showed that the obtained blend has improved properties compared to chitosan used alone. Multiple tests proved the occurrence of interactions in solutions containing the advantage of chitosan [160]. Chitosan/PVP blends may acquire a high ion exchange capacity [161]. To increase the physicochemical cytocompatibility between chitosan and PVA, Don et al. [162] proposed a two-step reaction to improve the cellular cytocompatibility of PVA. This is caused not only by the fact that chitosan interacts with cells, but also because it is accumulating on the top layer of membranes [163]. Chitosan/PVA blends exhibit similar antimicrobial properties to chitosan alone. Moreover, tested mixtures showed a reduction in edema because of increasing chitosan content [164]. Other studies by El-Hefian et al. presented an increasing level of hydrophilicity of obtained materials, again compared to chitosan alone [165].

#### 4.2.2. PEO Poly(Ethylene Oxide)

PEO is low-toxic, semicrystalline, bio-adhesive, mucoadhesive because of its water soluble synthetic polymer. Besides these features, it also possesses properties such as hydrophilicity, high viscosity, ability to form hydrogen bonds, and biocompatibility with other bioactive substances. Blends of chitosan/PEO were prepared to increase blood compatibility and permeability. Due to the combination of both polymers, higher hydration equilibrium than for chitosan alone was obtained. The observed increase in hydration of high porosity membranes was due to intermolecular associations between PEO and chitosan chains. The obtained blends offer opportunities to improve the permeability of toxic metabolites and reduce thrombogenicity in the case of hemodialysis [166]. According to Zivanovic et al., the addition of PEO to chitosan improves the physical, mechanical, and antimicrobial properties of the obtained films [167]. The addition of PEO, in the proportion of 75%, improved flexibility over three times in relation to chitosan alone. The addition of 25–50% PEO to chitosan films did not change the antimicrobial properties or any other functionality. Hence the conclusion that PEO added to the expensive chitosan material could reduce costs of biomaterial productions [168]. Chitosan/PEO blends are characterized by increased tensile strength and tear strength compared to pure chitosan. Moreover, the addition of PEO improves the flexibility of the obtained materials. According to Zivanovic et al., a mixture of chitosan:PEO 9:1 shows the best mechanical properties. The puncture strength of the film increased from 158.9 ± 17.0 N/mm for 50/50 chitosan:PEO blend ratio to 416.0 ± 24.7 N/mm for 100/0 chitosan/PEO blend ratio. Tensile strength values increased from 47.0 ± 8.8 MPa for 50/50 chitosan/PEO films to 73.5 ± 7.1 MPa for 100/0 chitosan/PEO films. 50/50 chitosan/PEO films showed a strong antibacterial activity with a reduction in viable cells of 3.27 log CFU/mL (Colony Forming Unit/ mililitre), which was similar to that of 100/0 chitosan/PEO films. This may be explained by the compatibility of chitosan/PEO blends. Moreover, chitosan/PEO blends do not crystallize and have high water vapor permeability [169]. Chitosan/PEO blends are soluble when the chitosan content does not exceed 50% (*w*/*v*). Moreover, the resulting mixtures may be amorphous, but only up to a PEO content of 20% [170].

#### 4.2.3. PVP Poly(Vinyl Pyrrolidone)

PVP is primarily a non-toxic, biocompatible polymer with excellent transparency and film-forming ability [171]. PVP has very good miscibility with chitosan mainly due to the hydrogen bonds formed as a result of the interactions between the carbonyl groups in the PVP pyrrolidone rings and the amino and hydroxyl groups in chitosan. Research by Sionkowska et al. showed the effect of UV radiation (254 nm) on various aspects of the obtained blends. After 4 h of irradiation, the level of mechanical changes of the obtained blends consisting of different polymer contents was almost constant, comparing to pure chitosan, in which the level of mechanical changes decreased by over two-fold. With an excess of PVP in the chitosan samples, changes of the mechanical parameters during irradiation are smaller than in the samples of pre-dominantly chitosan. This suggests that the presence of PVP in chitosan restrains its photochemical transformation. Moreover, the mixtures showed less susceptibility to photooxidation compared to PVP alone [160]. Additionally, chitosan/PVP blends may have a high ion exchange capacity [161].

#### 4.2.4. PNIPPAM Poly(N-Iso-propul-acrylamide)

Chitosan can be linked to PNIPPAM via traditional compounding or covalent bonding. PNIPPAM has no toxic properties. It is a biodegradable and biocompatible polymer and was originally combined with chitosan for drug delivery [172]. Ionic factors may have the chance to retool the molecular interactions between the polymer chains in the resulting mixtures. The adjustments in impacts cause changes in the properties of the polymer such as its swelling capacity, solubility, configuration, or amorphous/crystalline transition conditions. Two significantly improved features of the pre-pared chitosan/PNIPPAM blends are biodegradability and improved stimulus responses [173].

#### 4.2.5. PCL (Polycaprolactone)

PCL is a widely used synthetic aliphatic polyester with a semi-crystalline structure and a very high level of biodegradability and biocompatibility [174]. PCL has a low melting point (60 °C) and very high mechanical strength (elongation up to 1000% before break). Assessment of thermograms using Flory-Huggins theory, which showed a decrease in the melting temperature of PCL (from −2.738 Flory-Huggins interaction parameter to −8.274)) with an increase in chitosan content (0.25% to 0.75%), displayed very good miscibility of both polymers. The content of 75% PCL in the blend did not result in cytotoxicity. Moreover, the ratio of both polymers 1:1 increased the mechanical properties and the cellular support [175]. PCL degrades much slower than other biopolymers in human body fluids. This is due to the hydrolysis of the ester bonds which in turn leads to the formation of products such as carbon dioxide and water. Thanks to this property, polycaprolactone can be used in medicine in systems with a long biodegradation time (1–2 years) [176]. Moreover, numerous studies with positive results showing the high quality of cell proliferation on PLC-based scaffolds have been performed [177]. In turn, Yih-Lin Cheng et al. used polycaprolactone (PCL) as a base material. PCL was used because of its very good viscoelastic properties, high compatibility with other material mixtures, and inexpensive production. Despite many advantages of PCL, its unquestionable disadvantage is its hydrophobic nature that prevents the absorption of proteins and the attachment of cells to the surface of the material. PCL can be modified by adding hydrophilic polymers such as polyethylene glycol diacrylate (PEG-DA) to eliminate its hydrophobic properties. An example may be a quickly gelling at room temperature (23 °C) hydrogel (PEG-DA/chitosan) with a wide range of mechanical properties. It gels in the presence of a photo-initiator and UV light. Due to the addition of chitosan, much better adhesion properties of cells to the samples were obtained than in the case of samples without the addition of this biopolymer. SEM images clearly show that after 5 days of incubation, the chitosan/PCL samples present a significantly higher number of adhered cells when compared to the PCL-DA only sample due to the improved hydrophilicity of the surface and the biocompatible properties of chitosan. This can be inferred from the O.D values (570 nm) which are more than two times higher for the 15% chitosan samples as compared to the control group. Other advantages of combining these materials are the promotion of cell adhesion, cell differentiation, and proliferation. PCL has a low melting point, which results in easy formability and a thermally stable profile [178].

#### 4.2.6. PLA Polylactic Acid, or Polylactide

PLA and its copolymer composites show very good mechanical properties depending on the molar mass and polymer composition (from soft, flexible polymers to stiff and durable) [179]. It can be produced from renewable sources such as starch through a fermentation process. PLA is a thermoplastic, high-strength, high-modulus polymer and is considered biodegradable and compostable. Fimbeau et al. prepared a series of mixtures (10, 20, 30% PLA) and chitosan and subjected them to infrared spectra. Despite the good miscibility of both polymers, they did not observe any significant interactions between chitosan and PLA [180].

#### 4.2.7. PLLA Poly(L-Lactide)

Among many synthetic and biodegradable polymers, PLLA is distinguished by its extremely high biodegradability. In addition, it can be synthesized from renewable natural resources such as corn starch. It shows very good mechanical and thermal properties. Due to the increased strength, controlled biodegradability and improved chemical properties, resulting blends can be good candidates for the production of new biomaterials. In connection with the relatively cheap and easy production of PLLA, it can attest to a wider application compared to other polymers. FTIR and XPS (X-ray Photoelectron Spectroscopy) studies have shown interactions between PLLA carbonyls and the CS amino groups in the amorphous phase that result in the formation of hydrogen bonds. PLLA is a highly crystalline compound. After mixing it with chitosan its crystallinity is suppressed. However, the proper crystal structure does not change, and hydrogen bonds between the two polymers influence the crystallization of the mixtures. In turn, according to Chen et al. [181], the best chitosan:PEO blend ratio is 3:7.

#### 4.2.8. PAA Poly(Acrylamide)

PAA is a hydrophilic, high-molecular-weight synthetic polymer that likes chitosans-NH_2_ groups on the side chain and can form hydrogen bonds [182]. Desai et al. used a combination of chitosan and PAA in a ratio of 9:1. Such a blend gave promising properties. However, the described combination must be further investigated in the future [183].

### 4.3. Other Promising Substances

#### 4.3.1. Silica

It has been proven that the combination of chitosan with organic-inorganic composites can significantly improve chitosan activity. A large increase in the compressive strength (more than 10-fold) and the maximum strain (more than 25-fold) was achieved for a blend with 10% chitosan content [184]. Silica has many valuable properties for polymers. One of them is the tolerance to the growth of microorganisms as well as the large specific surface area [185]. Chitosan-silica-based composites may be a potential carrier for mono-, di-, and triphosphate nucleotides [186]. Reyes-Peces et al. obtained homogeneous chitosan-silica hybrid aerogels with chitosan contents up to 10 wt%, using 3-glycidoxypropyl tri-methoxy-silane as coupling agent, for tissue engineering applications.

#### 4.3.2. Montmorillonite (MMt)

The low mechanical strength of chitosan-based materials makes it necessary to combine it with other compounds or to improve its properties by crosslinking. MMt is a compound with a negatively charged surface and excellent cation exchangeability [187]. Chitosan can intercalate between MMt layers. According to Jafari et al. [188], montmorillonite influences the microstructure of hydrogels as well as their swelling and possible rate of drug release. High swelling of the hydrogels with the addition of MMt occurred under basic conditions (PH 7.4). In turn, a high rate of controlled release of sunitinib anticancer drug was observed at an acidic PH of 5.5. This was accomplished by ionic cross-linking of two biopolymers, κ-carrageenan, and chitosan, in the presence of magnetic montmorillonite nanoplatelets. Interestingly, it was observed that the amount of montmorillonite affected not only the microstructure of hydrogels, but also the drug loading efficiency of nanocomposite hydrogels were noticeably increased by introducing montmorillonite (from 69 to 96%).

## 5. Chitosan Derivatives

### 5.1. Carboxymethyl Chitosan CMC

The introduction of carboxyalkyl groups into the structure of chitosan as carboxymethyl is mainly carried out to increase the solubility level of chitosan. The reaction occurs either at the C6 hydroxyl group or at the NH_2_ moiety obtaining N-CMC, O-CMC, or N-O-CMC as products. CMC is an amphoteric and water-soluble chitosan derivative widely used in biomedicine. CMC as well as chitosan has excellent biocompatibility, biodegradability, and antioxidant activity. Therefore, these amphoteric polymers can be loaded with hydrophobic drugs and display strong bioactivity [189]. The properties of CMC closely depend on DS, which in turn depends on the amount of carboxylate agent and MW. Along with the MW increase, the value of DS (Degree of Substitution) decreases [190]. CMC has a very similar molecular structure to alginate [130] and agarose [191] in terms of carboxyl groups. On account of this property, both of these polymers have strong CMC compatibility. High miscibility was observed due to intermolecular interactions between CMC and alginate or agarose [192]. CMC has been crosslinked with alginate and agarose to use it as a scaffold for stem cell in situ differentiation into functional neurons and supporting neuroglia. With the maximum weight content of CMC 30% an improvement in the mechanical properties (from 10 to 13.80 cN/tex in the dry state) of the obtained fibers has been observed in comparison to pure compounds. The alteration of breaking elongation showed a tendency similar to that of the tensile strength, and the maximum value of 23.1% (in the dry state) was achieved when the CM-chitosan content was 10 wt% [130]. Liu et al. [193] proved that CMC can enhance the efficacy of the active constituents with poor solubility and bioavailability and at the same time increase brain drug concentration. In turn, Wahba et al. investigated the effect of CMC on Alzheimer’s disease. They proved that a CMC consisting layer delays the release of galanthamine and nanoceria in vitro [194]. The CMC derivative named methylated dimethyl-amino-benzyl chitosan (TM-Bz-CS) was tested by Hakimi et al. [195] in the context of Human Embryonic Kidney cells (Hek293). It was proved that TM-Bz-CS has significantly improved dissolution properties in a neutral, acidic, and alkaline environment, and does not exhibit cytotoxicity. It was evaluated on HEK293 cell line using XTT method with a result that cell viability brings 100%.

### 5.2. Acylated Modified Chitosan

The most common modification of the chitosan is acylation. It refers to the reaction of chitosan with a variety of organic acids and its derivatives by adding aliphatic or aromatic acyl groups to the molecular chitosan chain. The acylation reaction destroys the chitosan’s intermolecular and intramolecular hydrogen bonding which weakens its crystallinity and enhances water solubility.

There are two kinds of acylation: N-acylation and O-acylation. An acylation reaction that occurs with C2-NH_2_ forming an amide is called N-acylation [196]. The formation of the ester by the acylation reaction of C6-OH in the presence of a protective functional group on C2-NH_2_ is referred to as O-acylation [1]. N-acylated chitosan derivatives show enhanced biocompatibility, blood compatibility, and anti-coagulability. N-acylated chitosan can be used as a carrier or sustained-release agent as they do not cause an inflammatory reaction in the human body. The solubility of a described derivative depends on the degree of substitution and the length of the side chain. DS is proportional to the solubility. Additionally, a longer side chain results in higher crystallinity and a lower relative solubility. In turn, O-acylated chitosan exhibits different properties than O-acylated derivatives. O-acylation destroys hydrogen bond structures of chitosan while improving its fat solubility and hydrophobicity. An O-acylated derivative is lipid-soluble and can dissolve in non-polar solvents such as chloroform. It can be used as a stability biomaterial enhancer [189].

### 5.3. Quaternary Ammonium Chitosan

The quaternary ammonium is a positively charged hydrophilic group. Its addition to chitosan not only increases water solubility but also increases chargeability. Chitosan is positively charged at PH under 6.5, whereas quaternized chitosan is still permanently positively charged at PH above 6.5 The quaternization occurs with C2–NH_2_ and consists of introducing alkyl groups in place of the amino groups of chitosan. The most popular quaternization of chitosan products is N-Trimethyl Chitosan (TMC). It is considered to be one of the strongest existing mucoadhesive polymers. This is due to the presence of cationic groups in its chain [197]. Quaternary ammonium salt can be used for the quaternization reaction. It weakens hydrogen bonds and increases charging strength, which occurs with increasing water solubility. Quaternary ammonium chitosan salt has better antimicrobial, biocompatible, biodegradable, and non-toxic properties. It can penetrate mucus layers and bind to epithelial surfaces [198]. It is worth adding that the higher the DS degree, the better the water solubility properties and the higher the material potential [199].

### 5.4. Thiolated Chitosan

Thiolation is the reaction between primary amino groups of chitosan with coupling reagents that contain thiol groups [200]. Primarily, thiolated chitosan has a greater solubility in the aqueous environment but also has improved permeation and displays in situ gelling properties [201]. Thiolated chitosan derivatives have enhanced mucosal adhesion properties due to the formation of covalent bonds between free thiol groups and cysteine-containing glycoproteins in mucus [202]. The in situ gelling ability makes thiolated chitosan suitable not only for nose-to-brain applications but also for the elaboration of scaffolds. Not only chitosan but also metha-crylamide chitosan can be thiolated. As a result of such a reaction a porous, biodegradable material is created, which has an excellent effect on cell growth and neural stem target differentiation in 3D. The most effective method of making thiolated chitosan derivatives is the annealing method. Such a derivative has stronger adhesion, hydration ability, and drug release than other preparation methods [203].

### 5.5. Grafting Copolymerization of Chitosan-Polyethylene Glycol PEG

To achieve copolymerization of chitosan, it may be grafted onto it. A sample product of such reaction is polyethylene glycol (PEG)-grafted chitosan derivative which gives a significant increase in solubility over a wide PH range. Moreover, it shows increased muco-adhesion [204]. Other polymers that have been grafted to chitosan for Central Nervous System (CNS) application are gelatin, polylactic-co-glycolic acid (PLGA), poly (3,4 ethylene-dioxy-thiophene) (PEDOT), alginate, and agarose.

PEG is one of a limited number of biocompatible, synthetic polymers approved by the U.S. Food and Drug Administration (FDA) for biomedical applications [205]. PEG acrylates, such as PEG diacrylate (PEGDA), PEG dimethyl-lacrylate (PEGDMA), and multiarm PEG (n-PEG) acrylate (n-PEG-Acr) are the major macromers used for photopolymerization of tissue engineering scaffolds. PEG has been extensively used in hydrogels, as it is a biologically inert and nonimmunogenic polymer, which is known to confer greater water solubility. Injectable PEG-chitosan hydrogels have been developed by Kumar Sharma et al. for the controlled delivery of a ciprofloxacin antibiotic, and BSA, a model protein, for antibacterial applications. The MTT assay of PEG glyoxylic aldehyde showed that it exhibits negligible toxicity, with 98% cells viable at 0.1 mg/mL concentration, decreasing to 80% at 10 mg/mL concentration. Hydrogels with 12% PEG cross-linking were found to be most suitable for wound healing among the hydrogels investigated because of their highly efficient sustained-release properties [206]. Moreover, PEG acrylate hydrogels are attractive for use as resins in the 3D fabrication of scaffolds for regenerative medicine, although they are not naturally degradable, and show nontoxicity, non-immunogenicity, and aqueous solubility. Guiping Ma et al. [207], to modify chitosan. used the Michael reaction. This is the reaction of adding amines to unsaturated carbonyl compounds. In the case of chitosan, its chemical modification consists of adding monomers containing a double bond. The resulting biomaterial is a PEGDA-CS chitosan derivative. It can dissolve in distilled water and polymerize under UV light in the presence of an initiator. The compiled derivative is a biodegradable material with a wide range of applications. PEGDA-CS shows a high degree of crystallinity, completely different from the natural crystal structure of chitosan. Long chains of the compound hinder the formation of inter-molecular and extra-molecular hydrogen bonds. This results in the chitosan derivatives exhibit amorphous properties with a chaotic, disorderly arrangement of molecules [196,208]. This process leads to the loss of thermal stability by chitosan derivatives. It was observed that the antimicrobial activity against *Escherichia coli* was only slightly reduced with PEGDA-CS compared to pure chitosan [209]. Furthermore, prepared by Sadhasivam et al., chitosan blends with poly (EG-ran-PG) in a 1:1 ratio showed greater tensile properties than pure chitosan. Chitosan/P(EG-ran-PG) 1:1 blend demonstrates tensile strength at the level of 39 MPa, elongation at break 1.7%, and tensile modulus 45.8 MPa comparing to 24 MPa, 0.9%, and 37.8 MPa for pure chitosan, respectively. Blends showed no toxicity against mesenchymal stem cells. The chitosan/poly(EG-ran-PG) 1:1 blend samples not only ensure that the stem cells grow normally, but also promote their proliferation, which means that the blend films were biocompatible and bioactive. One possible reason for this could be the similarity between the structures of chitosan and glycosaminoglycans [210].

## 6. Chitosan as a Component of Biomaterials

Chitosan, due to its biological and physico-chemical properties, is commonly used as a component in a variety of biomaterials, such as membranes, hydrogels, scaffolds, particles and, lately, as a bio-ink component (Figure 4). The versatility of chitosan application is based on mechanical and biological material properties, accessible modifications, price, and easy manufacturing of desirable materials or systems.

We would like to highlight the possibility of chitosan nanostructures incorporation as a part of other materials, such as membranes, hydrogels, and scaffolds. Chitosan nanoparticles could play a significant role by increasing the endurance of biomaterial, improving biocompatibility, and elongating the time of biodegradation. It is worth highlighting that the chitosan nanoparticles could be used as nanocarriers, additionally releasing incorporated active substances, such as growth factors or specific drugs.

The following chapters fully describe the latest work on membranes, hydrogels, and scaffolds based on chitosan.

### 6.1. Chitosan Membranes

Chitosan, due to its biological and physico-chemical properties, is commonly used as membrane components that have a variety of applications, such as wound healing membranes [211,212,213,214,215], antibacterial membranes [216,217,218,219], membranes [220] and drug carrier systems [221,222,223].

Researchers have focused on the development of novel membrane material for chronic wound dressing applications to provide an optimum environment promoting the process of wound healing. Enumo et al. [224] developed a chitosan/pluronic membrane loaded with curcumin which exhibits, alongside tissue healing, antibacterial properties and a significant swelling ratio of up to 800%. Data showed that Minimum Inhibitory Concentration (MIC) for chitosan loading with curcumin is 25 mg/mL^−1^ and 100 mg mL^−1^ respectively to *S. epidermidis* and *P. aeruginosa*.

Chitosan blends with natural polymers, such as pectin, have gained popularity thanks to the accessibility and competitive material prices. Research conducting by Sari et al. [225] reports that a membrane composed with pectin isolated from *Cyclea Barbata Miers* and chitosan with immobilized extract with *Musa paradisiaca Linn* (Banana) accelerates wound healing by up to 86.9% in a 10 days experiment.

Blends of other natural polysaccharides, such as chitosan with *Bletilla Strata* polysaccharides, show antibacterial, antioxidant activity, and improvement of cell proliferation (experiment performed on mouse fibroblast: L929 cell line [226].

A PVA/chitosan membrane synthesized by solvent casting method, loading with propolis, showed remarkable results for wound healing [227]. The addition of propolis decreased membrane hydrophobicity by up to 50%, which is translated to better membrane adhesion to skin. The experiments performed on Mouse Embryonic Fibroblasts (MEF) showed an increased cell proliferation rate to 176 ± 13%, 775 ± 1%, and 853 ± 23%, at 24 h, 27 h, and 120 h, respectively.

Chitosan oligomer-electro-spun (COS) polycaprolactone exhibits antibacterial properties along with a greater concentration of chitosan oligomer [215]. Moreover, membrane induced the fastest haemostasis and affected wound healing in mice animal models. Membranes composed of chitosan are also finding applications as drug carriers. Jaisankar [228] found that chitosan blended with synthetic copolymer (thiourea, phenyl-hydrazine, and formaldehyde) exhibit promising kinetics of metformin releasing profile and its kinetics. Moreover, the membrane has been tested for antibacterial properties on selective bacterial and fungal strains. Obtained data implicate that the chitosan-copolymer blend inhibits microbial growth, exhibits wound healing properties, and could be potentially used as a drug delivery system for diabetic patients. Ilk et al. report [229] that natural chitosan membrane obtained from insect’s corneal lenses also found application as drug delivery systems. Chitosan derivatives are also popular among membranes dedicated to drug systems. Ziegler-Borowska [230] claim, that chitosan-based drug delivery systems can improve the targeting of porphyrinoids and their release at predetermined locations and finally achieve desired therapeutic effects with minimal side effects. Moreover, the chitosan/porphyrinoid combinations have revealed enormous benefits for antimicrobial photodynamic therapy.

### 6.2. Chitosan-Based Scaffolds

Chitosan is also well known as an adjunct to scaffolds dedicated to soft tissue, cartilage, ligament and bone tissue regeneration. Combining chitosan with other materials such as gelatin [231], alginate [232], β-tricalcium phosphate [233,234] bioactive glass [235], hyaluronic acid [236] or silk fibroin [237] scaffold acquired desirable properties, enhancing tissue regeneration by cell proliferation and vascularization.

Shen et al. [231] studied the chitosan and gelatin porous scaffolds, covered by dipping in polylactic acid to provide better mechanical properties. The pore size of the scaffold varies from 30 µm to 100 µm and can be controlled. A degradation study conducted in physiological pH reveals that the loss mass of scaffolds after 7 days of incubation oscillated at around 35%. However, the hydrolysis of the system in the presence of lysozyme indicated that chitosan can inhibit the pH fluctuation caused by lactic acid throughout the process of polylactic acid degradation.

Zhensheng et al. proved that [232] the proliferation of Human Chondrosarcoma Cells was faster on the chitosan-alginate scaffolds than on pure chitosan. Moreover, after two weeks of study, cells seeded on chitosan scaffold were showing fibroblast-like morphology, whereas the cells seeded on a chitosan-alginate scaffold presented spherical morphology. It is worth highlighting that SDS-PAGE electrophoresis reveals the presence of collagen II type, characteristic of chondrocytic phenotype. Those findings suggest that a chitosan-alginate scaffold could be suitable for cartilage tissue engineering.

Chitosan scaffolds reinforced by β-tricalcium phosphate (β-TCP) and calcium phosphate invert glass exhibit microporous structure which could be achieved by adjustment of β-TCP- glass ration, and their ratio to chitosan content [233]. During the incubation of chitosan and chitosan-glass scaffold in SBF (Simulated Body Fluid) solution, no appetite was formed. However, the addition of β-TCP yields apatite formation, which suggests scaffold bioactivity. Moreover, the author claims that the apatite layer was expected to be osteoconductive and resorbable, which is crucial for the bone tissue biomaterials.

Scaffolds composed of chitosan and β-TCP were also the subject of a study by Yin et al. [238]. The researchers, instead of glass, used gelatin as a scaffold component to obtain a microporous scaffold with diverse pore sizes. The biocompatibility study was performed on rabbits. The mild inflammatory response was observed after 12 weeks of inculcation. Nevertheless, the obtained results suggest the utility of these components in non-loading bone regeneration.

Chitosan is also applied in scaffolds dedicated to ligament tissue regeneration. Funakoshi et al. [239] developed chitosan, chitosan-based and hyaluronan hybrid fibres by the wet spinning method. The authors found that a layer of hyaluronan covered chitosan fibres significantly increase the number of attached fibroblasts. Moreover, results obtained via SEM indicate that fibroblast culture developed collagen fibres after 14 days from seeding. The tensile strength measured on produced fibres during a degradation study in standard medium revealed that, after 28 days of incubation, the tensile strength decreased from over 200 MPa (1 day) to an estimated 75 MPa. The DNA content was also studied during the 28 days. The results show a significant increase in the amount of DNA from 40 µg/sample to over 100 µg/sample after 4 weeks of the experiment. The obtained results were in agreement with the chitosan fibres, and two variations of chitosan fibres covered by hyaluronan. The data derived from the study suggest that chitosan fibres can be a promising material for ligament tissue engineering.

Chitosan and chitosan- poly(lactide-co-glycolide) (PLGA) fibres were studied as potential cartilage tissue scaffolds [240]. The chitosan-PLGA fibres exhibit unique soft and strong mechanical properties which are not observed in these materials separately. The chondrocytes proliferation on scaffolds increased twofold from 5 to 20 days of incubation. Moreover, the presence of type II collagen must be highlighted as characteristic for cartilage tissue.

In his work, Nezhad-Mokhtari [241] presented chitosan-based hydrogel scaffolds fortified with golden nanoparticles dedicated to soft tissue engineering. Results obtained show that injectable collagen/nanocrystalline cellulose/chitosan loaded with golden nanoparticles scaffold do not reveal cytotoxicity on NIH-3T3 (cell line of highly contact-inhibited cells established from NIH Swiss mouse embryo cultures) fibroblast cell line. The mechanical properties of gel could be easily adjusted by changing the weight ratio between materials by adding cellulose or chitosan to the reinforced scaffold structure.

Perez-Puyana claims, that chitosan has the potential to become the new collagen applied in genipin-crosslinked scaffolds [242]. Research was conducted on raw chitosan and collagen materials and the influence of genipin was evaluated. The best results were obtained for the system with a 2% chitosan concentration.

Jiang [243] used 3D printing to print a PLA scaffold with the net of the channel to provide better diffusion of substances inside the construct. The printed scaffold was immersed into chitosan solution, freeze-dried and prepared for further modifications. The author claims that the integrated bilayer scaffolds with the structural parameters in each layer were fabricated and exhibited no delamination. The fibroblast cells seeded on the construct showed nearly 100% cell viability after 7 days of incubations.

Chitosan-based scaffolds combined with Neural Stem Cells (NSC) are also applied in spinal cord injury [244]. The researcher observed the creation of neurofilament between the scaffold and host tissue, which is promising for long term study.

Chitosan, thanks to its properties, is often used as an extender or to improve mechanical and rheological properties, such in Garakani [245]. Combining cartilage Extracellular Matrix (ECM) with chitosan and agarose, Garakani obtained a novel system with adequate properties for nasal cartilage tissue engineering application.

Bombaldi de Souza [246] combined phosphorylated chitosan with xanthan gum to boost osteoinduction in scaffolds dedicated to bone engineering. Despite that the material is not toxic, it does not exhibit proper roughness, porosity and degradation time, and the cell adhesion and proliferation need to be improved by further surface modification.

The other scaffold composition dedicated to bone tissue engineering was proposed by Parfizivard [247]. The proposed scaffold was composed of poly (3-hydroxybutyrate) PHB-Chitosan (Cs)/multi-walled carbon nanotubes (MWCNTs) nanocomposite coating deposited on nano-bio-glass (nBG)-titania (nTiO_2_) scaffolds fabricated by foam replication method. The results indicate a lack of scaffold cytotoxicity, favourable cell attachment further proliferation and perfect mechanical properties.

### 6.3. Hydrogels

Hydrogels can provide an environment similar to hydrated tissue. Due to this ability they play an important role in tissue engineering. Hydrogels can also be designed as drug carriers or scaffolds [16]. Similarly to previous cases, chitosan is often combined with other promising substances such as gelatin [248,249], collagen [250] or graphene oxide [251] to modify and optimize hydrogel properties.

Chitosan hydrogels are mostly dedicated to skin wound healing [252,253,254,255,256,257,258,259], but also find application as biomaterial applied to bone regeneration [260], spinal cord regeneration [261] or heart disease [262].

Zhang et al. [262] combined catechol functionalized chitosan with oyster peptide microspheres (OPM) and β-sodium glycerophosphate (β-GP) to obtain thermosensitive hydrogel for skin wound healing. According to the results, the described hydrogel accelerates fibroblast migration, and accelerates collagen production and creation of new vessels around the wound. Moreover, the authors noticed increased production of total protein (TP) which indicates a faster regeneration process.

Hydrogel consisting of chitosan, nano-hydroxyapatite and tilapia peptides was described by Qianqian et al. [263] and dedicated to burn skin wound healing. The experiments were performed on rabbits. The results suggest that hydrogel decreased inflammatory reaction, and accelerated wound healing by increasing vascularization, collagen production and expression of VEGF. Similar results were obtained for the system consisting of chitosan-ulvan hydrogel with the addition of nanocrystal cellulose [264].

The hydrogel constructed with oxidized chitosan and amidated pectin design by Amirian et al. [253] exhibited interesting swelling properties, gelation time and degradability. Despite the chemical description, biological results are required for complete system characterization.

The hydrogel phase is one of the main phases in the development of bio-ink. Given the popularity of chitosan hydrogels as the main component or additive to bio-ink, the data describing the latest trends in chitosan hydrogels are described in more detail in the Section 7. The latest achievements of chitosan application as nanoparticle carrier material is vastly described in Table 1.

### 6.4. Chitosan Microparticles and Nanoparticles as Drug Carriers

A drug delivery system is defined as a technology focused on the delivery and management of drug release kinetically for active agents. The variety of drugs, active substances, and forms constantly require the search for new solutions to provide the delivery of the active substance. Among the various delivery systems, nanoparticles [266,270,271,273,274] and microparticles [269,272] are the most recognized and popular forms of carriers.

The recently described systems consisting of chitosan as main or additive material are shown in Table 1. In some cases, chitosan particles can increase Drug Loading Efficiency (DLE) [266,269,270,271,273,274] and Drug Encapsulation Efficiency (DEF) [266,269,274], which is a crucial parameter in drug delivery systems.

## 7. Chitosan as a Bio-Ink Composition Material

Bio-ink is defined as a material used for 3D printing. Bio-ink consists of living cells and biomaterials that mimic the extracellular matrix environment, supporting cell adhesion, proliferation, and differentiation after printing. In terms of biomaterials, bio-ink requires all the properties of classic biomaterials, such as biocompatibility and non-toxicity. Taking into account the advisability of 3D printed materials, bio-inks must cover the following requirements: (i) printing temperatures adequate for cells’ temperature range; (ii) mild cross-linking or gelation conditions; (iii) non-toxic bioactive components which allow further material structure modification by the cells after printing.

The most common bio-ink materials are alginate, ECM, collagen, gelatin, agarose, and hyaluronic acid. Chitosan, due to its physico-chemical and mechanical properties, is gaining popularity as an addition to bio-inks [275,276].

Chitosan bio-ink has been applied in bioprinting of artificial organs and structures in the human body such as cartilage tissue [277], bone tissue [278], neural connections [279], liver or heart valves [280,281], which shows the versatile use of the material in the bioprinting process [Figure 5]. Table 2 presents the current trends for chitosan application as a bioink composition.

## 8. Limitations and Future Prospective in Biomedicine

Chitosan is considered as an attractive biomaterial for regenerative medicine, buy it also possesses certain limitations. The major challenge is a lack of a general harmony in analyzing various physicochemical properties of chitosan such as molecular weight or degree of acetylation, as these properties are directly related to safety concerns. An example is that LMW chitosan creates a more precise shape than HMW, therefore it is suggested for use in order to achieve more dimensional accuracy. One of the consecutive factors that preclude the application of chitosan is the inter and intra-molecular hydrogen bonding of a rigid crystalline structure that limits its solubility. Pure chitosan does not support proper cell adhesion at a sufficiently high level, therefore needs to be modified to simplify this process.

Furthermore, it decomposes at a temperature above 220 °C which is limiting to the high-temperature modification process. Additionally, a strong limitation is unsatisfactory mechanical properties, due to which chitosan is often combined with other materials to obtain sufficient qualities. Moreover, blending chitosan with other materials can lead to improvement in printability, fidelity, or even the creation of novel cell-laden matrixes. Moreover, the chitosan bio-ink can be blended with other materials that possess the desired chemical, physical and biological properties according to the final biomaterial application, such as nanocellulose [285], calcium phosphate [284], formic acid and lactic acid [283].

The featured review summarizes different strategies that have been used to overcome these conditions, e.g., a wide variety of chitosan derivatives. Nevertheless, there is still a lack of knowledge about the relation between molecular changes and the acquired biological properties of these derivatives. It has been demonstrated that special care should be taken in the clinical use of chitosan over a long time due to the possible disturbances in intestinal microbial flora, fat-soluble vitamins, mineral absorption and bone mineral content deficit [277,278].

Knowledge of chitosan-based biomaterials is required for further investigations. In particular, more research is required to comprehensively investigate the toxicity of chitosan and its derivatives to human beings and other living organisms. Environmental protection and green production in the development of chitosan-based products for applications in various fields for the benefit of humans must also be considered [195,282,283,284].

Despite all the described limitations, chitosan offers desirable properties compared to other biomaterials. CS, as a part of any material, could introduce valuable properties such as antimicrobial activity, mucoadhesive, and biocompatibility, which are in demand for biomedical use.

Additionally, chitosan derivatives seem to be even more attractive. Its properties could be improved by chemical modification to reach more suitable properties for biomedical material. The number of patents filed over the last decade shows the increasing popularity of chitosan in biomedicine. Most of these were granted for the healing of wounds, hemostatic products and tissue engineering. Many of the chitosan products are reported as hydrogel beads as an intra-articular supplement, gene therapeutic agent, novel injectable delivery system for stem cell therapy, scaffolds that can replace autogenous bone, osteogenesis inductor, nanocarriers dedicated for drug delivery systems, bioprinting, and tissue dressing materials. Innovative chitosan application as a component of biotopes is bringing promising results and opens new possibilities for regenerative and personal medicine. Chitosan and its derivatives will exhibit broader prospects in biomedicine in the future [211,285]. By combining organs, 3D bioprinting with nanomaterials as a bio-ink component, with the addition of active substances (drug, VEGF factor, other desired substances) can reach a new level of personalized medicine, with organs tailored for specific patients.

Thanks to its features, chitosan can be successfully considered as a biomaterial of the future. Chitosan in nanoscale is mostly applied as a drug carrier system. Chitosan gold-based nanoparticles with a size range of 67 nm–196 nm are dedicated to doxorubicin delivery [266]. Low molecular weight chitosan-cyanocobalamin particles with a diameter of around 85 nm are applied in the oral delivery of ciprofloxacin [271]. Silk-coated chitosan-gold nanoparticles with a size of between 3 nm–8 nm were developed to target-directed antitumor agents [274]. Those examples of chitosan nanoparticle application show further potential possibilities of chitosan use.

An interesting approach would be to use chitosan nanospheres with incorporated VEGF factor (Vascular endothelial growth factor) to enrich bio-ink and simplify vascular growth inside 3D printed structures. Moreover, thanks to the modification of chitosan nanospheres, for example by the layer-by-layer method, the possibility exists of controlling incorporated substances’ release from nanoparticles. Due to this approach, increased mechanical and biological properties of bio-ink could be attained along with controlled substance release.

## 9. Conclusions

The review brings new insights into known chitosan applications. Chitosan possesses desirable chemical, physical and biological properties as a biomaterial. It is worth pointing out that this is a review of the latest literature reports mostly available over 2 years (2020–2021), which sheds new light on the possibilities of chitosan application in medicine. The review provides convenient access to a significant amount of recent data concerning chitosan-based biomaterials and their applications. The data exhibited in the tables include parameters of the obtained structures and basic chemical and/or biological characterization.

The crucial seventh review section emphasizes chitosan-based bio-inks in cutting-edge 3D organ bioprinting technology. This ground-breaking method in combination with a well-known natural biomaterial allowing the achievement of remarkable results in the regenerative medicine field, equally in soft and hard tissue. The potential application of this well-known polysaccharide is increasing, in parallel with many new varieties of structure synthesized in nanoscale, which is expanding chitosan application in the biomaterials field.

Many studies have confirmed the versatility of chitosan application both at the micro and nano scale. Thanks to the analysis of the chitosan enriched systems introduced in the literature, new applications can address existing problems through the use of innovative solutions in the biomaterials field.

## Figures and Tables

**Figure 1 nanomaterials-11-03019-f001:**
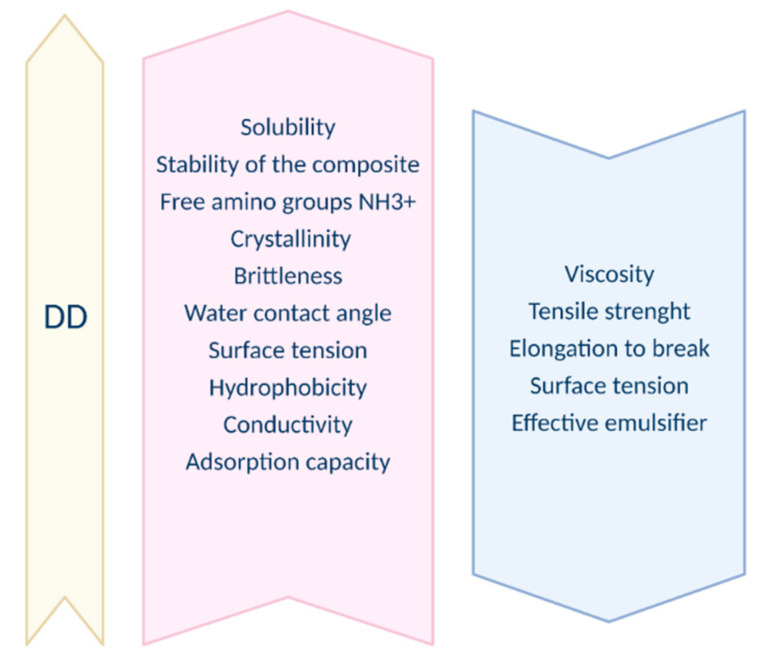
DD influence on physical and chemical properties.

**Figure 2 nanomaterials-11-03019-f002:**
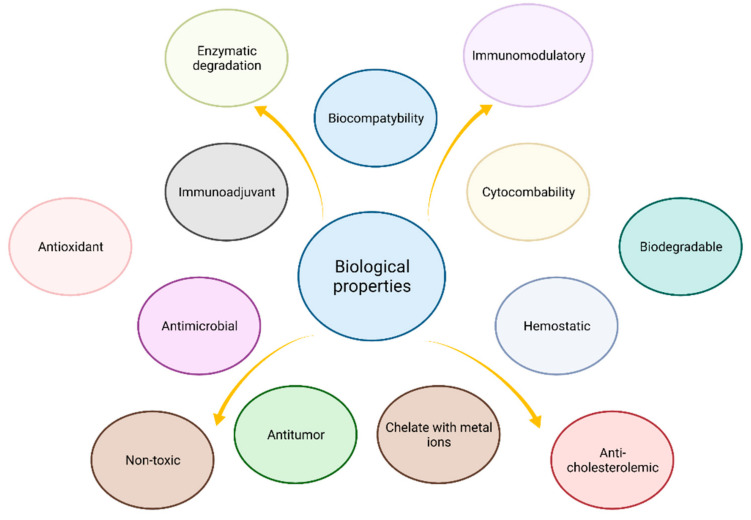
Biological properties of chitosan.

**Figure 3 nanomaterials-11-03019-f003:**
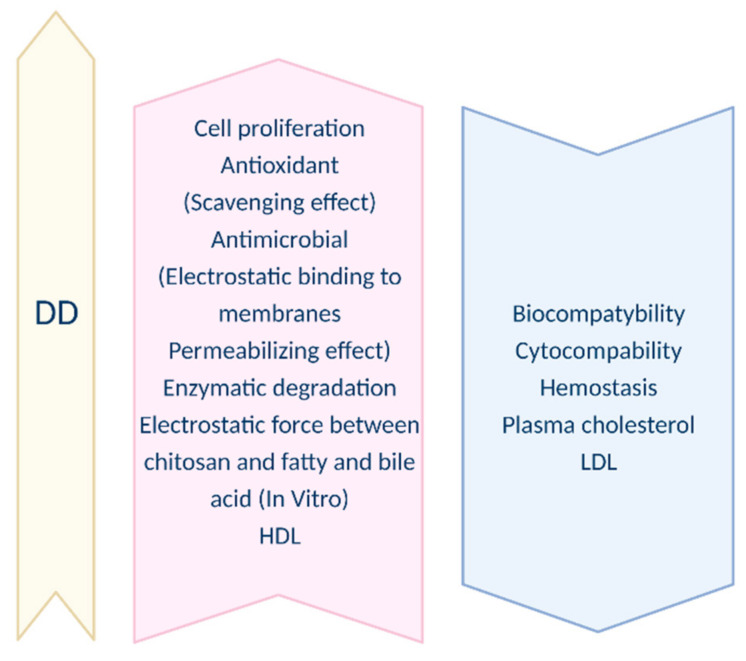
DD influence on biological properties.

**Figure 4 nanomaterials-11-03019-f004:**
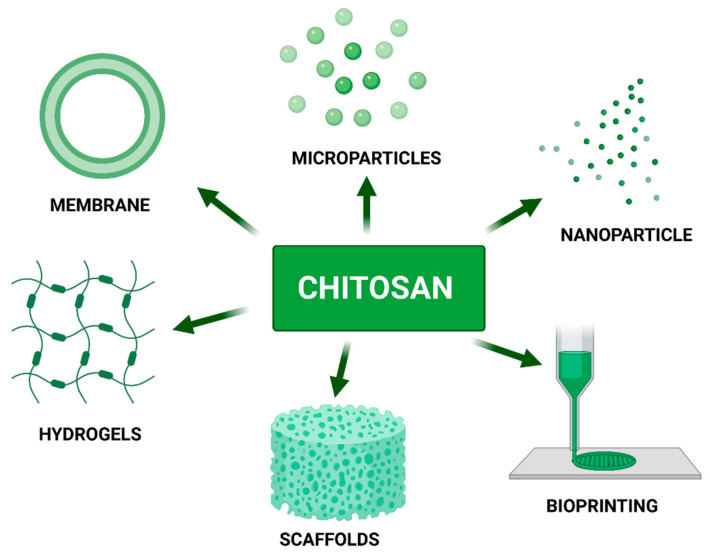
Chitosan application as a material dedicated to medicine.

**Figure 5 nanomaterials-11-03019-f005:**
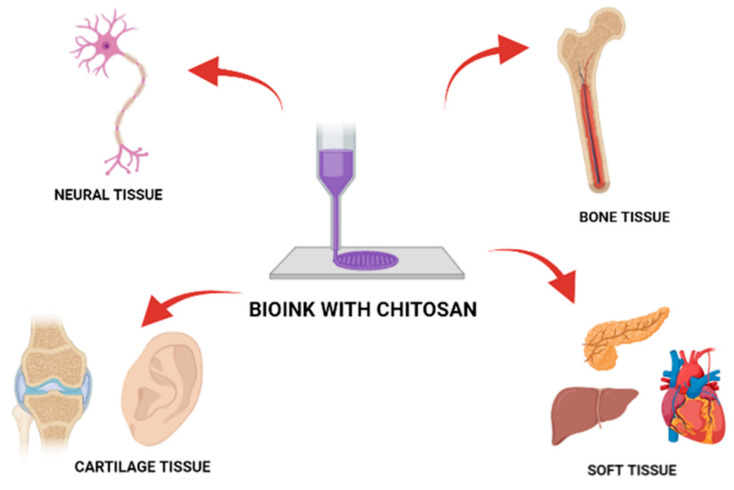
Application of chitosan bio-ink as an material for regenerative medicine.

**Table 1 nanomaterials-11-03019-t001:** Drug delivery systems based on chitosan.

Component	Aim/System Dedication	Cytotoxity/Cell Proliferation	Biodegradability/ Swelling Ratio	Drug Loading Efficiency (DLE)/Drug Encapsulation Efficiency (DEE)	Drug Release	Comments	Bibliography
Pullulan oxidation with TEMPO/NaBr/NaClO (PT) Pullulan oxidation using sodium periodate (PP) Chitosan (C)	Drug delivery	Lack of information	37 °C in PBS, 24 h–168 h, after 7 cycles of drying-rehydration swelling degree was over 500% for each beads type (CPT, CPP, C)	As a test compound was used ibuprofen, neomycin, and bacitracin: 95–85% drug release efficiency (from loaded)	Ibuprofen release profile: 4–6 h to reach plateau	Tunable release behaviour, antibacterial activity	[265]
Chitosan-gold based aptamer consisting of bio-synthesized gold nanoparticles (Au NPs), chitosan (CS) with aptamer (Apt)	Drug delivery system of 5-fluorouracil (5FU) and doxorubicin (Dox) to improve glioblastoma treatment	Glioblastoma cell line (LN229); Fluorescent microscope, cytometry	Lack of information	DEF: 75–96% DLE: 1.3–11% for various formulations of nanoparticle	pH 5.4 for 5FU 90% in 120 h, for DOX 55% in 120 h: pH 7.4 for 5FU 40% in 120 h, for DOX 15% in 120 h	Average particle size: 67 nm–196 nm	[266]
(Acrylic acid)-co-(2-acrylamido-2-methylpropane-sulfonic acid) (AAc/AMPS)	Colon cancer drug delivery	Lack of information	At pH 7:8 g/g in 300 min; at pH 1:2.5 g/g in 300 min	Lack of information	96% of 5-FU drug at pH 7 after 7 h	Incorporation of Cs with AAc and AMPS forming terpolymer hydrogel	[267]
Chitosan-coated magnetic alginate (CMAM)	Drug release under mild condition	Lack of information	Lack of information	Lack of information	77% in 420 min at pH 7.4 and 10% at pH 2.1	The smaller particles (220 μm) show a faster release rate than the bigger materials (1000 μm)	[268]
Magnetic microspheres based on pectin coated by chitosan (mag-Pec-Cs) Metamimzole (MTZ) as a drug model	Multi-responsive drug delivery system	Lack of information	pH 1.2: 275% and 200% for magnetic and pectin covered particles; pH 6.8: 110% and 60% for pectin covered particles;	Pec-Cs/MTZ, DEF% and DLE% 85 ± 1% and 0.14 ± 0.02%, while mag-Pec-Cs/MTZ DEF% and DLE% equal to 88 ± 2% and 0.15 ± 0.04%,	At pH 6.8, the 75% MTZ released after 12 h.	The release process can be adjusted by varying the pH of the medium	[269]
Lactobionic acid-modified thymine-chitosan nanoparticles (Thy-Cs NPs)	Carriers for methotrexate delivery (MTX)	HepG2 cells, MTT test; cell viability close to 90% for particle concentration rage 31–1000 µg/mL non-loaded particles	Lack of information	DLE (Drug Loading Efficiency) around 63–90%	Thy-Cs/MTX NP at pH 7.4, 6.5 and 5.5 release in 120 h incubation 90%, 60% and 20% incorporated MTT respectively	Size about 190–250 nm; growth inhibition in three-dimensional multicellular tumour spheroids.	[270]
Low molecular weight chitosan-cyanocobalamin nanoparticles (LMCSCNbl)	Oral delivery of ciprofloxacin (CIP)	HEK 293 cell lines, 24 h MTT test 18, 9, and 1% cytotoxicity for LMCS-CNCbl at concentrations of 100, 10, and 1 μg/mL,	Lack of information	DLE 57%	Drug release close to 80% in 24 h, pH 7.4 and 0.1 M HCl	Average diameter 85 nm	[271]
Chitosan-ferulic acid microcapsules (CF) loaded with BSA	Oral carrier in functional foods and drug delivery systems.	Lack of information	Swelling ratio: CS and CS-FA (Chitosan-Feluric Acid) after 400 min: 120% and 160% respectively	Encapsulation efficiency: 64–78%;	Drug accumulated release of CS-FA microspheres: after 14 h: 55%, 48% and 46% was noticed for 0.15 g, 0.2 g and 0.1g, of drug	The chitosan-ferulic acid conjugates exhibited low crystallinity but high thermal stability compared with that of chitosan.	[272]
Phenylboronic acid-conjugated chitosan nanoparticles	Curcumin delivery for tumor treatment	HepG2 cells, MTT 24 h test: 90%, for phenylboronic acid-conjugated chitosan nanoparticles	Lack of information	DLE in range: 40–90% for curcumin	Cumulative release: pH 7.4, 5.5, and 5.5 and mM H_2_O_2_	Spherical shape, size 200–230 nm; NPs exhibited efficient antitumor efficiency against cancer cells	[273]
Silk-coated chitosan-gold nanoparticles (CAu)	Target-directed delivery of antitumor agents	HeLa cell line: 100%, CAu-DOX was 40% and 30% for 0.25, 0.5 and 1.5 µM concentration	Lack of information	The DLE in NPs was 70.83%, and the drug encapsulation efficiency of the beads was calculated to be 87.45%.	Coated and uncoated nanoparticles: 60% and 90% during 32 h, respectively	Nanoparticle size 8 + 3 nm; beads size: 900–1000 µm	[274]

**Table 2 nanomaterials-11-03019-t002:** Novel bio-inks for 3D printings based on chitosan.

Chitosan Blends	Material Dedication	Rheology	Toxicity/ Cell Proliferation	Biodegradability Swelling Ratio	Comments	Bibliography
Carboxymethyl Chitosan-Based Bioink: Ethylenediaminetetraacetic acid (EDTA) stabilized with 0.5 M calcium chloride	Cartilage tissue	Storage modulus G′ at 23 °C: 112 kPa	Rabbit chondrocytes; flow cytometry; 95:9 ± 1:3% After 36 h seeded on mesh; similar proliferation rate between the control group (9:9 ± 0:7%)	Swelling ratio: 14–22% weight increase after 22 days in water	Bioprinting of scaffolds for cells	[277]
Cell-Laden Thermosensitive Chitosan Hydrogel Bioink: β-glycerophosphate Potassium phosphate Sodium bicarbonate	Development of chitosan-based bioink	Storage modulus G′ at 36 °C: around 1000 Pa	Human periodontal ligament stem cells; WST (Colorimetric assay for the nonradioactive quantification of cell proliferation, cell viability, and cytotoxicity )assay showed that there was no significant difference in cell viability until day five	Lack of information	Cell encapsulation is associated with minimal cytotoxicity	[278]
Cell-laden hydrogels, bioink: Potato starch	3D bioprinting scaffolds for neural cell growth		Neuro-2a, mouse neuroblastoma cells LDH (Lactate Dehydrogenase)assay kit and fluorescent microscopy: viability after 10 days-10% and lower	Degradation time is decreasing with the addition of the potato starch component	Chitosan dissolution and crosslinking must be optimized	[279]
BMSCs-laden gelatin/sodium alginate/carboxymethyl chitosan hydrogel: Gelatin Sodium alginate Carboxymethyl chitosan	3D bioink for tissue scaffolds	Young modulus: 80–120 mPa	Bone mesenchymal stem cells (BMSC); Live/Dead cells staining: 85% of the printed cells were viable at 0 and 2 days of culturing	Biodegradation in 60 days in the physiological environment: 35–50% mass loss	Bioink showing antimicrobial properties towards *E.coli*	[280]
Fabrication of hydroxybutyl chitosan/oxidized chondroitin sulfate hydrogels: Hydroxybutyl chitosan (HBC) Oxidized chondroitin sulfate via Shift base reaction	Cell delivery system for cartilage tissue engineering	Turn into stabile hydrogel at 35 °C–40 °C, Storage modulus G′: −150–300 Pa for 50 mg/mL HBC concentration	Mesenchymal stem cells, Live/ Dead assay after via fluorescent microscopy; Cell viability was verified inside the hydrogels in 14 days, showing gradually spreading in the hydrogel with the appearance of pseudopodia	Lack of information	Injectable hydrogel with a porous structure of average 100 µm pore size was developed to form a microporous hydrogel	[281]
DLP printing photocurable chitosan: Methacrylic anhydride Lithium phenyl-2,4,6-trimethylbenzoylphosphinate (LAP)	Photocurable bioink for digital light processing (DLP) technology for tissue engineering	Stress-strain for CHIMA (methacrylated derivative of chitosan) 33.6% 80 kPa; CHIMA 44.6% 33.6% −150 kPa	Human umbilical vein endothelial cells (HUVECs); LIVE/DEAD Viability/Cytotoxicity kit: Viability for 4 examined samples oscillated around 90% after 3 days from incubation	The swelling ratios of hydrogels 11.7–33.6% DS exhibit a decreased trend from 500% to 150% 2 during the incubation time	The CHI-MA (1 wt%) with 33.6% DS was selected as the photocuring bioink for DLP	[282]
Photocurable chitosan as bioink Methacrylated chitosan Β-glycerol phosphate salt (β-GP)	Bioink for cellularized therapies towards personalized scaffold architecture	40 s of exposition at 37 °C initiate crosslinking bioink: Storage modulus G’: −90–100 Pa	NIH, 3T3, Saos-2, SH-SY5Y cell lines; LIVE/DEAD Viability/Cytotoxicity kit, fluorescence microscopy: viability: around 95–115% compering to control, after 24 h	Decreased mass of 55% after 14 days of incubation in the cultured medium at 37 °C; thermogravimetric analysis	Bioink did not adversely affect the hosting cells and allowed cell proliferation and organization towards tissue formation.	[196]
Chitosan ducts fabricated by extrusion-based 3D printing Formic acid Acetic acid Glycolic acid Lactic acid	Soft tissue restoration	Young modulus: 12.38 ± 1.19 MPa	MTT test on L929 mouse fibroblast cell line for 24 h cell viability of CS ducts prepared by 30% GA close to 90%	Stable after soaking in two weeks in Tris-HCl with the addition of lysozyme	The 30 wt.% GA was optimal based on tensile properties and preliminary cytotoxicity	[283]
Chitosan-calcium phosphate inks: Calcium Phosphate Acetic acid orthophosphate solutions	Bioinks as potential bone substitute	For all other inks, Loss modulus G″ were higher than G′ from the start, thus the inks were liquid-like	Not tested	Lack of information	More printable inks are obtained with higher chitosan concentration (0.19 mol·L^−1^).	[284]
Cell-Laden Nanocellulose/Chitosan-Based Bioinks: Glycerophosphate Hydroxyethyl cellulose Cellulose nanocrystals	Bioprinting and enhancing cell differentiation for bone tissue	Viscosity in the range of 30 Pa·s–6 × 10^4^ Pa·s; Yield stress 412.35 ± 45.35 pa	MC3T3, a pre-osteoblast cell line; Live/Dead cell staining kit; after 7 days incubation in media at 37 °C there is neither significant proliferation nor cell toxicity	The shrinkage of scaffolds after 24 h incubation in DMEM (Dulbecco’s Modified Eagle Medium) at 37 °C ranges between 30–34%		[285]
Natural based poly(gamma-glutamic acid)/Chitosan bioink: Poly(gamma-glutamic acid)	Alternative to other materials used in 3D bioprinting	Storage modulus G′ around 50 Pa and 30 Pa for 4.5% and 6% Chitosan hydrogels	Human adult fibroblast: Cell viability after 14 days incubation of DMEM on bioink around 80%	35% Mass loss after 35 days incubation in cell medium	FTIR analysis demonstrated Gamma-PGA/Cs interpolyelectrolyte complex formation	[286]
A writable bioink under serum culture media: Catechol Vanadyl ions	Polymer for 3D printing	Storge modulus G′ value of the V-Chi-C gels at 1 Hz was gradually enhanced up to 6 ± 0.5 × 10^6^ Pa at 168 hrs from 69 ± 18 Pa at 0 hr	LIVE⁄DEAD^®^ Viability/Cytotoxicity Kit; 90% L929 cells viability after 5 days incubation on scaffolds	Weight loss down to 50% of initial mass at 12 h of incubation in PBS, and remained constant (40%) for next 7 days		[281]

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
