# Peer review of "Chitosan as an Underrated Polymer in Modern Tissue Engineering"

_nanomaterials, 2021, doi:10.3390/nano11113019_

Round 1
Reviewer 1 Report
The review is well done and there is some few relevant novelties. However, in my opinion this review should not published for two main reason:
1) This journal is about nanomaterials and the review is centered in a macropolymer.
2) there is a high number of reviews on chitosan and in my opinion despite the up-to-date information there is no need to another review on chitosan at the moment.
Reviewer 2 Report
This review provides a summary of chitosan and its modification. It can give readers some guide. However, I still have some concerns listed below. Please modify it before acceptance.
- Figures 1 - 4 also have the logo in the lower left corner. I think it can be removed unless it is required by the software.
- The critical input by the authors and any explanation on future prospects based on the authors experience is missing.
- Conclusion is too short. The authors should add perspectives.
- Why is there so little content in section 6.2? And there is no reference?
- There are many errors regarding “space”. It should be improved.
Author Response
Dear Reviewer,
We would like to sincerely thank you for reviewing our manuscript “Chitosan as an underrated polymer in modern tissue engineering”. Your suggestions significantly improved our work. We revised the manuscript according to your suggestions. Below we present answers. All included changes were made according to your suggestions. Everywhere where it was necessary, an appropriate comment was included.
Thank you very much for your time while making this revision. We are looking forward to your next opinion.
Sincerely,
Marta Klak, PhD

Reviewer 3 Report
The Authors here presented an interesting review that demonstrates the successful use of chitosan and chitosan-based biomaterials in tissue engineering.
The Authors covered several aspects of the use of chitosan aimed for medical applications. In my opinion the overall description resulted quite generic and more significant details about results should be added to the text to better understand the behavior of chitosan before to accept the paper for publication.
Conclusions are too generic. It would be interesting to add at the end a paragraph describing some future perspectives of the use of chitosan in medical applications resuming clearly advantages, limitations and innovative aspects.
Author Response

(The authors gave the same response as above.)

Round 2
Reviewer 1 Report
Despite the authors explanation the review remain, in my opinion, out of focus in term of nanotechnology. However, I recognize the effort of the authors in highlight the connection to nanomaterials in some part of the review. Thus, I recommend the authors to focus the review only in the nanotechnology application of chitosan reformulating all the review removing all the non-relevant parts.
Author Response
We are deeply thankful for the time and effort you put to review our work again. According to your suggestion, the non-related chapters to the nanomaterials topic were deleted and the article was modified. Again, we would like to express our gratitude for revising our article.
Please see the attachment

Reviewer 3 Report
After Author revision, the paper is now suitable for publication in Nanomaterials.
Author Response
We are deeply thankful for your time and effort to review our publication.
Round 3
Reviewer 1 Report
I recognize the effort of the authors to improve the manuscript